# Local axonal morphology guides the topography of interneuron myelination in mouse and human neocortex

Jeffrey Stedehouder[1], Demi Brizee[1], Johan A Slotman[2], Maria Pascual-Garcia[1], Megan L Leyrer[3], Bibi LJ Bouwen[4,5], Clemens MF Dirven[5], Zhenyu Gao[4], David M Berson[3], Adriaan B Houtsmuller[2], Steven A Kushner[1]*

[1]Department of Psychiatry, Erasmus MC University Medical Center, Rotterdam, Netherlands; [2]Erasmus Optical Imaging Center, Department of Pathology, Erasmus MC University Medical Center, Rotterdam, Netherlands; [3]Department of Neuroscience, Brown University, Providence, United States; [4]Department of Neuroscience, Erasmus MC University Medical Center, Rotterdam, Netherlands; [5]Department of Neurosurgery, Erasmus MC University Medical Center, Rotterdam, Netherlands

*For correspondence:
s.kushner@erasmusmc.nl

**Competing interests:** The authors declare that no competing interests exist.

**Abstract** GABAergic fast-spiking parvalbumin-positive (PV) interneurons are frequently myelinated in the cerebral cortex. However, the factors governing the topography of cortical interneuron myelination remain incompletely understood. Here, we report that segmental myelination along neocortical interneuron axons is strongly predicted by the joint combination of interbranch distance and local axon caliber. Enlargement of PV+ interneurons increased axonal myelination, while reduced cell size led to decreased myelination. Next, we considered regular-spiking SOM+ cells, which normally have relatively shorter interbranch distances and thinner axon diameters than PV+ cells, and are rarely myelinated. Consistent with the importance of axonal morphology for guiding interneuron myelination, enlargement of SOM+ cell size dramatically increased the frequency of myelinated axonal segments. Lastly, we confirm that these findings also extend to human neocortex by quantifying interneuron axonal myelination from ex vivo surgical tissue. Together, these findings establish a predictive model of neocortical GABAergic interneuron myelination determined by local axonal morphology.

## Introduction

Myelination is the insulating ensheathment of axons by oligodendrocytes to enhance action potential propagation and provide metabolic support (*Simons and Nave, 2015*). Recent studies have shown that a large fraction of neocortical myelination arises from axons of fast-spiking, parvalbumin-positive (PV) interneurons (*Micheva et al., 2016*; *Stedehouder et al., 2017*). Nearly every cortical PV+ interneuron is myelinated, and most frequently with a proximally-biased axonal topography consisting of short internodes interspersed with branch points (*Stedehouder et al., 2017*; *Stedehouder et al., 2018*). In contrast, other neocortical GABAergic interneuron subtypes are more rarely and sparsely myelinated, raising the question of what factors determine this cell type-restricted pattern of neocortical interneuron myelination.

Axonal diameter has been previously demonstrated as an important neuronal factor influencing myelination. In the peripheral nervous system (PNS), a critical threshold of axonal diameter of ~1 µm has been identified, which largely predicts myelination (*Duncan, 1934*). However in the central nervous system (CNS), the pattern is much less clear. Axons with diameters ~200 nm can become myelinated (*Hildebrand et al., 1993*), while axon diameters as large as ~800 nm can remain

unmyelinated (*Hildebrand et al., 1993*). In vitro, oligodendrocytes reliably initiate myelination of synthetic nanofibers above a critical diameter of ~300 nm but rarely do so for smaller diameters (*Lee et al., 2012a*; *Bechler et al., 2015*; *Goebbels et al., 2017*). The diameter findings appear to extend to grey matter in vivo, where myelinated axons <300 nm are rarely observed (*Micheva et al., 2016*; *Goebbels et al., 2017*). Moreover, following acute demyelination regenerated myelin sheaths often re-establish their pre-morbid pattern of myelination, suggesting that intrinsic axonal factors are primary determinants of myelination (*Auer et al., 2018*). Furthermore, oligodendrocytes appear to sense axonal diameters in vitro, and adjust their internode length based on fiber diameter (*Bechler et al., 2015*). Taken together, axonal diameter is firmly established as an important determinant underlying internode formation. However, in contrast to its high predictive validity in the PNS, axonal diameter is only moderately predictive of myelination topography in the CNS.

Here we examine the relationship between cortical interneuron myelination and axonal morphology in adult mouse prefrontal cortex. We find that the topography of myelination along individual PV+ axons is strongly predicted by the joint combination of axonal diameter and interbranch distance. The bivariate model combining axonal diameter and interbranch distance was superior to univariate models involving either axonal diameter or interbranch distance alone. We further explored the model robustness by implementing bidirectional manipulations of PV+ interneuron size. Enlargement of PV+ interneuron size resulting from cell-type specific deletion of *Tsc1* increased the incidence of myelinated segments. Conversely, reduction of PV+ interneuron size by cell-type specific deletion of *Ube3a* decreased the frequency of myelinated segments. Yet notably, in both cases, the joint combination of interbranch distance and local axon caliber remained highly predictive of myelin topography. Lastly, we considered regular-spiking SOM+ cells, which normally have relatively shorter interbranch distances and thinner axon diameters than PV+ cells, and are rarely myelinated. However, enlargement of SOM+ cell size by cell type-specific deletion of *Tsc1* dramatically increased the frequency of myelinated axonal segments and with a topography accurately predicted by the bivariate model. Lastly, we find that interneurons reconstructed from human ex vivo surgical tissue also exhibit similar rules governing their axonal myelination. Together, these results establish a highly predictive model of neocortical GABAergic interneuron myelination topography based on local axonal morphology.

## Results

### Super-resolution imaging of individual fast-spiking, PV+ interneuron axons

To examine the relationship between the axonal morphology of PV+ interneurons and their myelination, we targeted fluorescent PV+ interneurons in the adult medial prefrontal cortex (mPFC) of *Pvalb*::cre (*Hippenmeyer et al., 2005*),Ai14 (*Madisen et al., 2012*) mice for whole-cell patch-clamp recording and biocytin filling ($n = 8$ cells; *Figure 1a,b*; *Supplementary file 1*). Recorded cells exhibited the fast-spiking pattern associated with PV+ interneurons (*Figure 1c*). Biocytin-labeled cells were imaged by confocal microscopy for reconstruction (*Figure 1d*), followed by structured illumination microscopy (SIM) for high-resolution analysis of individual axonal segments (*Chéreau et al., 2017*) (see Materials and methods; *Figure 1—figure supplements 1–3*). We systematically analyzed PV+ interneuron axonal segments up to the 7th branch order, beyond which myelination was rarely observed in this region (*Stedehouder et al., 2017*; *Stedehouder et al., 2018*). Axon shaft diameter averaged $0.34 \pm 0.01$ μm (range 0.16–0.98 μm) and decreased with increasing branch order (*Figure 1e–g*). *En passant* boutons, located primarily on more distal branches ($\geq$5th branch order), averaged $0.71 \pm 0.01$ μm in diameter (range 0.34–1.26 μm; *Figure 1h*).

### PV+ interneuron myelination co-varies with axon morphology

Myelin was visualized by immunolabeling for myelin basic protein (MBP). Consistent with previous studies (*Stedehouder et al., 2017*; *Stedehouder et al., 2018*), reconstructed PV+ interneurons consistently exhibited myelination of their proximal axons (8 out of 8; 100%, *Figure 2a,c*), while distal axonal segments remained unmyelinated (*Figure 2b,c*). Myelination of PV+ interneurons typically extended from branch point to branch point (*Figure 2c*), in which ~84% of myelinated internodes had their boundaries within 5 μm of an axonal branch point (*Figure 2d*). For each reconstructed

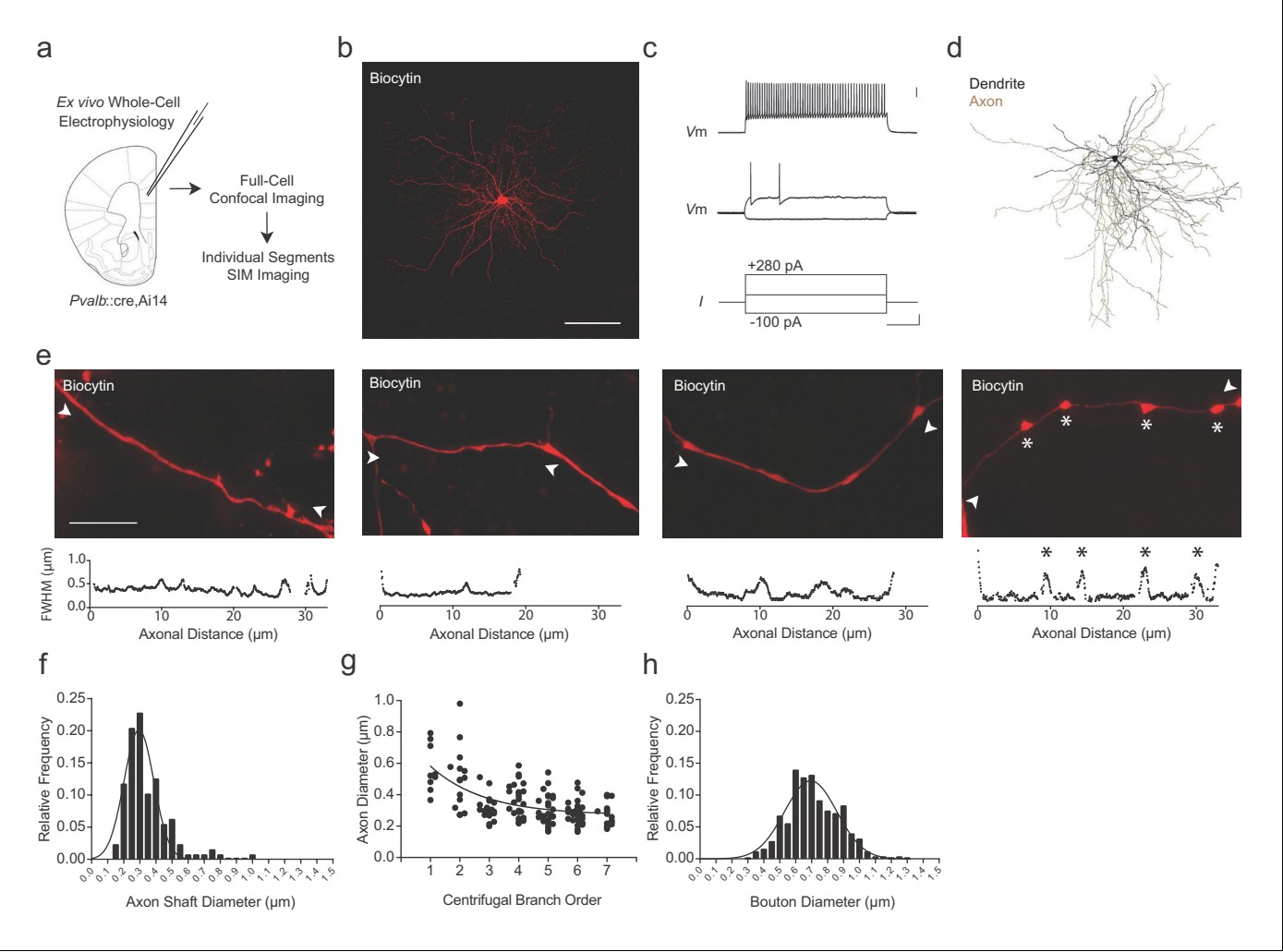

**Figure 1.** Super-resolution microscopy of fast-spiking, PV+ interneuron axons. (**a**) Experimental approach. Biocytin-filled fast-spiking PV+ interneurons from mPFC were analyzed using both confocal imaging and structured illumination microscopy (SIM) imaging. See also *Figure 1—figure supplements 1–3*. (**b**) Maximum projection confocal image of a representative biocytin-filled PV+ cell from mPFC layer V (red). Scale bar, 50 μm. (**c**) Current clamp recording of evoked action potentials. Scale bars are 20 mV, 100 pA and 100 ms from top to bottom (right). (**d**) Full reconstruction of a mPFC layer V PV + interneuron. Soma and dendrites in black, axon in brown. (**e**) Representative SIM *z*-stack projections of PV+ interneuron axonal segments (top), along with their corresponding FWHM diameter profiles (bottom). White arrowheads indicate measurement boundaries. From left to right: First branch order axon initial segment; second branch order unmyelinated axonal segment; third branch order myelinated axonal segment; sixth branch order unmyelinated axonal segment featuring multiple *en-passant* boutons (indicated by asterisks). Scale bar, 10 μm. (**f**) Distribution histogram of PV+ interneuron axon shaft diameters, fitted with a Gaussian curve. *n* = 140 axonal segments/8 cells. (**g**) Average axon shaft diameter decreases steadily over centrifugal branch order. *n* = 140 segments/8 cells. p<0.001, one-way ANOVA. (**h**) Distribution of axonal *en passant* bouton diameters of PV+ interneuron axons, fitted with a Gaussian curve. *n* = 250 boutons/8 cells. Abbreviations: FWHM, full-width half-maximum. I, input current. SIM, structured illumination microscopy. Vm, membrane voltage.

The online version of this article includes the following source data and figure supplement(s) for figure 1:

**Source data 1.** Diameter measurements for axonal segments (f), branch order (g), and en passant boutons (h) of PV::WT cells.
**Figure supplement 1.** Experimental flowchart for axonal reconstructions.
**Figure supplement 2.** Axonal diameter analysis.
**Figure supplement 3.** Locations of biocytin-filled and reconstructed PV+ and SOM+ cells.

interbranch segment, we examined the relationship between the probability of segmental myelination and the average axon shaft diameter using receiver operating characteristic (ROC) analysis (*Hanley and McNeil, 1982*). Axon shaft diameter strongly co-varied with myelination, with a critical univariate threshold at 334 nm (area under curve, AUC = 0.93; sensitivity = 97.2%, specificity = 84.4%), above which internodes were often present and below which myelination was rarely

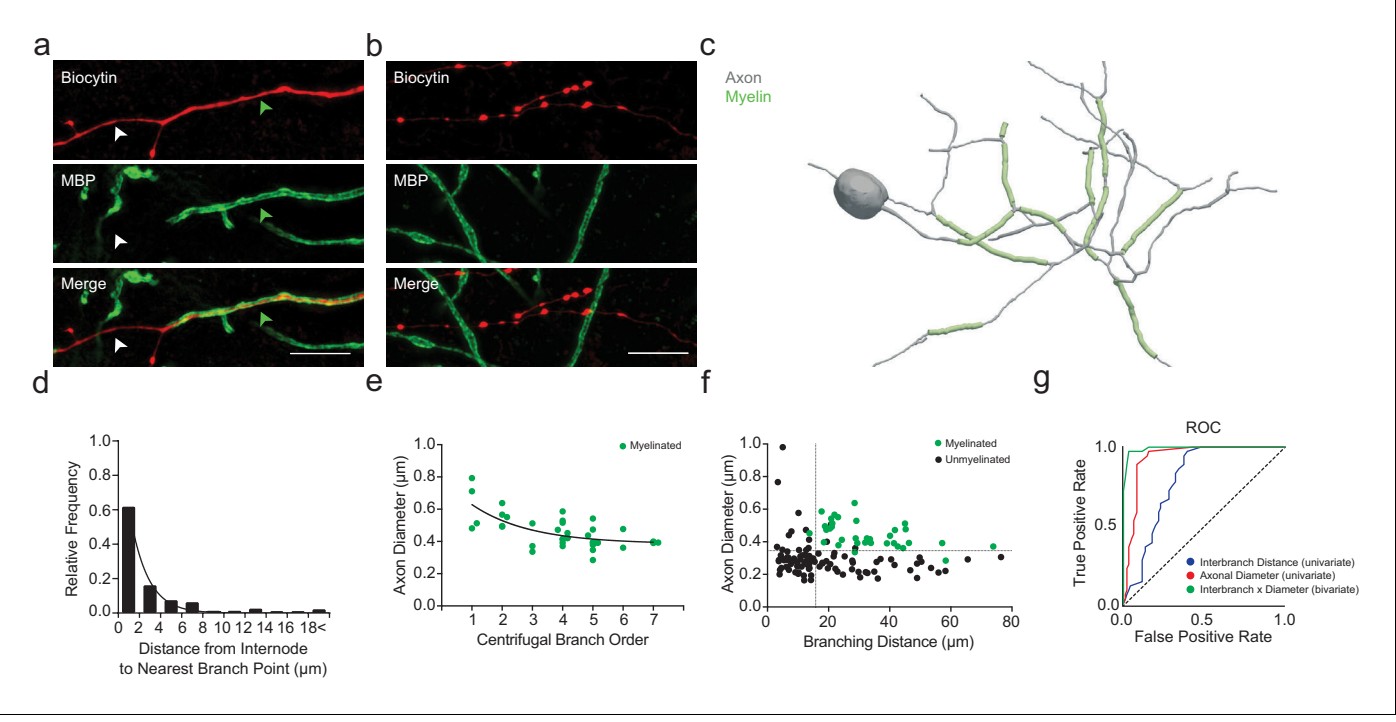

**Figure 2.** PV+ interneuron axon diameter co-varies with myelination. (**a**) Average SIM *z*-stack projection of a biocytin-filled PV+ interneuron axon (red) along with its myelination (MBP; green arrowhead), centered over a fourth branch order segment. Note that the relatively thinner axonal segment is unmyelinated (white arrowhead). Scale bar, 5 µm. (**b**) Average SIM *z*-stack projection of a biocytin-filled PV+ interneuron axon (red) lacking myelination (MBP), centered over a seventh branch order segment. Note the frequent *en passant* boutons and thin axon shaft. (**c**) Neurolucida reconstruction of an mPFC fast-spiking PV+ interneuron axon. Axon in grey, myelinated segments in green. Note the proximal onset of myelin, consisting of short internodes interspersed by branch points. (**d**) Frequency histogram of nearest neighbor distance from internodes to branch points. *n* = 81 segments/5 cells. (**e**) Average axon segment diameter versus branch order, exclusively for segments showing myelination. *n* = 39 segments/8 cells. p<0.001, one-way ANOVA. (**f**) The joint combination of axonal diameter and interbranch point distance is highly predictive of segmental myelination. Each circle represents an individual axonal segment. Myelinated segments (green) are consistently thicker and longer compared to unmyelinated segments (black), with critical thresholds (dotted lines) of 13.4 µm and 334 nm for interbranch distance and diameter, respectively. *n* = 140 segments/8 cells. (**g**) Receiver-operator characteristic (ROC) curves for interbranch distance (blue) and diameter (red) as univariate predictors, as well as the significantly improved joint bivariate prediction (green) of myelination status (p<0.001). Diagonal dotted line indicates the non-discrimination reference boundary. Abbreviations: ROC, receiver-operator characteristic.

The online version of this article includes the following source data for figure 2:

**Source data 1.** Morphological measures in PV::WT cells: internode-to-branch point (d), branch order (e), and bivariate interbranch distance / axonal diameter values for myelinated and unmyelinated segments (f).

observed (*Figure 2e–g*). However, 30% of interbranch segments with axon shaft diameter >334 nm remained incorrectly classified by the univariate ROC model (15 of 50 segments), suggesting additional deterministic factors underlying the variance in segmental myelination.

We next considered the univariate relationship of interbranch distance with segmental myelination. We found that myelination occurred more frequently along uninterrupted interbranch segments greater than 17.4 µm (AUC = 0.79; sensitivity = 97.2%, specificity = 60.4%; *Figure 2f,g*). However, similar to the relationship with axonal diameter, a substantial proportion of interbranch segments >17.4 µm were unmyelinated (48%, 35 of 73 segments).

Therefore, we implemented a bivariate ROC analysis (*Jin and Lu, 2009*) to explore whether the intersection of interbranch distance and axonal diameter might yield improved estimates of segmental myelination. Using a bivariate ROC analysis, the optimal interbranch distance and axonal diameter were 13.7 µm and 334 nm, respectively (AUC = 0.99; *Figure 2f,g*). The joint combination of these two thresholds correctly predicted whether 128 of 132 segments contained an internode (97.0% accuracy), a significant improvement over the univariate models (interbranch distance: 70.5% accuracy, Fisher's Exact Test p<0.001; axonal diameter: 87.9% accuracy, Fisher's Exact Test

p=0.005). In particular, the bivariate model correctly predicted 35 of 36 myelinated internodes (sensitivity = 97.2%), matching that of the univariate predictors (Fisher's Exact Test p=0.99). However, the joint combination of interbranch distance and axonal diameter also predicted with high accuracy those segments in which myelination was absent (93 of 96 segments, specificity = 96.9%), in contrast to the univariate models (Fisher's Exact Test; axonal diameter: p=0.004, interbranch distance: p<0.001). These findings suggest that the combination of interbranch distance and axonal diameter are highly predictive of segmental myelination along PV+ interneurons.

To independently corroborate the findings from supra-resolution imaging, we utilized electron microscopy (EM) for assessing the morphology and myelination of PV+ cell axons. We utilized a genetic labeling method to enhance the electron dense contrast of PV+ cell axons by stereotactic injection of a cre-dependent adeno-associated virus (AAV2) into the mPFC of *Pvalb*::cre mice. Virus-transduced *Pvalb*::cre cells expressed the EM marker APEX2 (*Karube et al., 2004*) fused with membrane-targeted GFP (mGFP-APEX2), permitting cell-type specific visualization of labeled PV+ cells in both fluorescence and electron micrographs (*Figure 3a–c,e,f*). Prior to ultrastructural analysis, we used confocal microscopy to confirm that GFP+ neurons resembled PV+ cells morphologically. We also confirmed that nearly all of the virally labeled cells were immunopositive for PV (93.9 ± 1.3%;

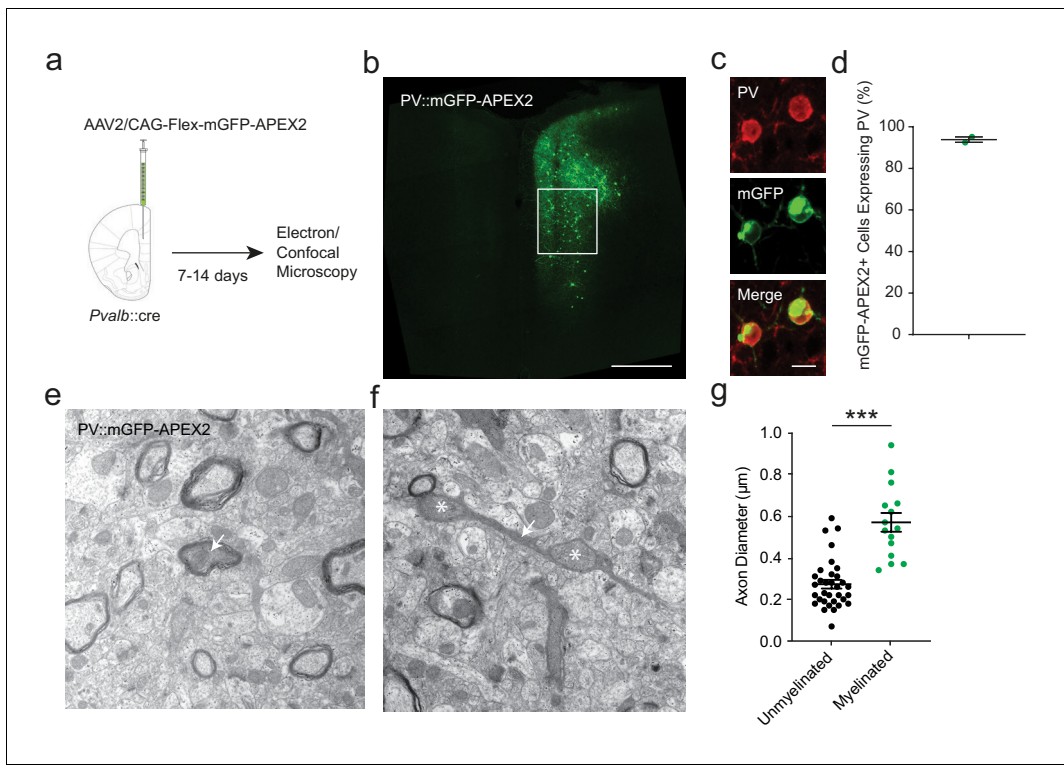

**Figure 3.** APEX2 contrast-enhanced electron microscopy confirms that smaller-diameter axonal segments lack myelination. (a) Experimental flowchart. *Pvalb*::cre mice were given unilateral injections into mPFC with AAV2/CAG-flex-mGFP-APEX2, and sacrificed for confocal and electron microscopy after 7 to 14 days. (b) Confocal image of unilateral PV-specific mGFP-APEX2 expression in mPFC (green). White square depicts the region of interest in the prelimbic area. (c) Representative confocal microscopy image of mGFP-APEX2 fluorescence (green) and its colocalization with PV immunofluorescence (red). (d) Quantification of colocalization between mGFP-APEX2+ cells and PV immunofluorescence. 93.9 ± 1.3% of mGFP-APEX2+ cells expressed PV. *n* = 2 mice (e–f) Electron microscopy images (14kx) of PV::mGFP-APEX2+ myelinated (e) and unmyelinated (f) axons (arrows). Morphological enlargements featuring mitochondria (asterisks) were not included in the diameter analysis. (g) PV+ interneuron axonal segments featuring myelination have a larger average diameter (green) than unmyelinated axons (black). Unmyelinated: 0.269 ± 0.019 μm, *n* = 38; myelinated: 0.570 ± 0.045 μm, *n* = 15. ***p<0.001. Unpaired two-tailed Student's *t*-test. Black bars represent mean ± s.e.m.

The online version of this article includes the following source data for figure 3:

**Source data 1.** Quantification of PV::WT cell axonal diameter by APEX2 contrast-enhanced electron microscopy (g).

*Figure 3b–d*). We then used APEX2-mediated peroxidase histochemistry to produce intracellular electron dense labeling. PV+ cells were readily detected by their darkened cytoplasmic staining in electron micrographs. This allowed us to identify axonal processes belonging to PV+ interneurons, whether myelinated or not (*Figure 3e,f*). The distributions of axon shaft diameter and bouton diameter, and their relationship with axonal branch order, were also highly comparable to previous ultrastructural analyses of fast-spiking, PV+ interneurons (*Karube et al., 2004*; *Nörenberg et al., 2010*; *Hu and Jonas, 2014*). Consistent with the fluorescence microscopy analysis (*Figure 2*), myelination was observed exclusively around PV+ axonal fibers with shaft diameters > 330 nm. Mean diameter was significantly larger in myelinated than in unmyelinated segments (*Figure 3g*). Thus, axon morphology strongly predicts myelination of PV+ interneurons using two independent methods.

## Bi-directional manipulation of PV+ axon morphology alters myelination

In order to examine the robustness of axonal morphology in predicting PV+ interneuron myelination topography, we performed cell-type specific manipulations known to influence axonal geometry. Deletion of the *Tsc1* gene has been previously shown to induce enlarged somata of various neuronal cell types across a diversity of brain regions (*Fu et al., 2012*; *Normand et al., 2013*; *Meikle et al., 2007*; *Carson et al., 2012*). Moreover, the Akt-mTOR pathway, a downstream target of *Tsc1*, is one of the main regulators of axon caliber (*Markus et al., 2002*). Conversely, mice harboring a deletion of *Ube3a* have recently been shown to exhibit smaller neurons (*Sidorov et al., 2018*; *Wallace et al., 2012*) with reduced axonal diameters in corpus callosum (*Judson et al., 2017*).

To obtain PV cell-specific deletions, *Pvalb*::cre mice were crossed with floxed *Tsc1*$^{f/f}$ mice (PV:: TSC1) and floxed *Ube3a*$^{p/f}$ mice (PV::UBE3A) (*Figure 4a*; *Figure 4—figure supplements 1–2*). PV+ cells in adult mPFC of PV::TSC1 mice exhibited a ~50% increase in soma size, in accordance with a strong upregulation of pS6$^{235/236}$, a downstream target of mTOR (*Figure 4b,c*). PV::TSC1 cells showed filopodia-like extensions on their soma and proximal dendrite, which were not observed in PV::WT cells (*Figure 4—figure supplement 1f*). Conversely, PV::UBE3A mice exhibited a ~15% reduction in PV+ interneuron soma area (*Figure 4b,c*). Notably, mPFC PV cell density was similar across PV::UBE3A, PV::TSC1, and PV::WT mice (*Figure 4—figure supplement 3a–b*).

To examine axon caliber, adult PV::TSC1, PV::UBE3A and PV::WT mice received unilateral stereotactic injections in the mPFC of adeno-associated virus (AAV) containing cre-dependent GFP. Fourteen days later, mice were sacrificed and axons originating from GFP+ somata were imaged using SIM (*Figure 4d*). Since we observed that the diameter of the 1$^{st}$ branch order strongly correlated to diameter of consecutive axonal segments (*Figure 1*), we measured primary axonal diameter at the 1$^{st}$ branch order for GFP-labelled cells from PV::TSC1, PV::UBE3A and PV::WT cells as a high-throughput indication of axon caliber. Consistent with enlarged somata, this analysis revealed a significantly increased primary axonal caliber of PV::TSC1 PV+ cells compared to PV::WT cells (*Figure 4e,f*). In contrast, axons from PV::UBE3A showed a non-significant trend toward decreased primary axon caliber compared to PV::WT cells (*Figure 4e,f*).

We next performed whole-cell electrophysiological recordings of adult mPFC PV+ cells combined with biocytin-filling and *post hoc* MBP immunofluorescence as previously described (*Figure 5a*; *Figure 4—figure supplement 4*) (*Stedehouder et al., 2017*; *Stedehouder et al., 2018*). Recorded PV+ cells in PV::TSC1, PV::UBE3A and PV::WT exhibited fast-spiking firing characteristics, with no detectable differences between genotypes (*Figure 5b–d*; *Figure 4—figure supplement 4*). Confocal imaging followed by proximal axonal reconstructions (*Figure 5e–h*) revealed that all PV::TSC1 cells exhibited axonal myelination (13 of 13 cells; 100%; *Figure 5f–i*), similar to PV::WT cells (10 of 11 cells; 90%) and PV::UBE3A cells (7 of 8 cells; 88%). *Tsc1*-deficient PV+ cells showed increased axonal myelination per cell compared to PV::WT cells (*Figure 5j*). Interestingly, the increase in myelination per cell was associated with both an increase of average internode length (*Figure 5k*) as well as a higher number of internodes per cell (*Figure 5l*), but without a change of interbranch distance (*Figure 5m*). Onset of myelination, measured as axonal distance from the soma or originating dendrite to the beginning of the first internode, was unchanged (*Figure 5n*). In contrast to PV::TSC1 cells, *Ube3a*-deficient PV+ cells exhibited a decrease in axonal myelination (*Figure 5i,j*), including both a lower number and shorter length of internodes (*Figure 5k,l*), consistent with their reduced soma size. Finally, the initial point of myelin onset from the soma was unchanged in PV::UBE3A cells (*Figure 5n*). Despite these PV+ cell type-specific alterations in myelination, no robust changes were

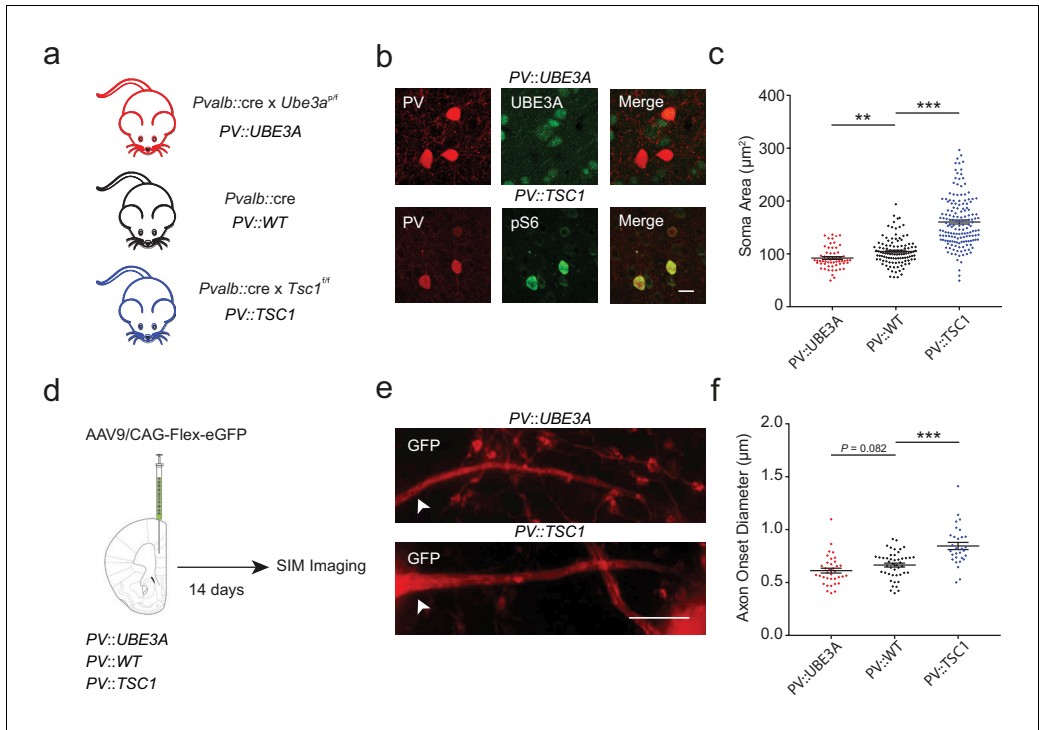

**Figure 4.** PV::TSC1 and PV::UBE3A mice exhibit reciprocal alterations of PV+ cell morphology. (**a**) Overview of mouse breeding scheme. (**b**) Maximum projection confocal image of PV+ somata (red) in PV::UBE3A mice lacking UBE3A (green; top), and in PV::TSC1 mice showing high pS6 expression (green, bottom). Higher magnification images are provided in *Figure 4—figure supplements 1–2*. Scale bar, 15 μm. (**c**) Quantification of PV+ interneuron maximum projection soma area from mPFC layers II-V. PV::UBE3A: 92.1 ± 2.7 μm$^2$; *n* = 58 cells; PV:: WT: 104.6 ± 2.5 μm$^2$; *n* = 109 cells PV::TSC1: 160.2 ± 3.8 μm$^2$; *n* = 159 cells; *n* = 3 mice per group. (**d**) Experimental procedure. PV::UBE3A, PV::WT and PV::UBE3A were stereotactically injected with AAV9/CAG-Flex-eGFP in the mPFC and analyzed two weeks later with SIM imaging. (**e**) Representative projection of a SIM *z*-stack showing primary axon branches (red) originating from transfected GFP+ somata in PV::UBE3A and PV::TSC1 mice. White arrowhead indicates location of axon onset. Scale bar, 2 μm. (**f**) PV+ interneuron axon caliber quantifications from mPFC layers II-V. PV::UBE3A: 0.613 ± 0.023 μm; *n* = 36 axons; PV::WT: 0.665 ± 0.019 μm; *n* = 46 axons; PV::TSC1: pS6+ 0.846 ± 0.034 μm; *n* = 30 axons; *n* = 3 mice per group. ***p<0.001; **p<0.01; *p<0.05. One-way ANOVA followed by *post hoc* Tukey's test. Black bars represent mean ± s.e.m.
The online version of this article includes the following source data and figure supplement(s) for figure 4:

**Source data 1.** Soma area (c) and axon onset diameter (f) for PV::UBE3A, PV:WT, and PV::TSC1 cells.
**Figure supplement 1.** PV::TSC1 mice exhibit PV-specific deletion of *Tsc1*.
**Figure supplement 2.** PV::UBE3A mice exhibit PV+ cell-specific deletion of *Ube3a*.
**Figure supplement 3.** PV cell-specific mutations of *Ube3a* or *Tsc1* do not alter PV+ cell density in mPFC.
**Figure supplement 4.** Electrophysiological properties of PV::TSC1, PV::WT, and PV::UBE3A cells.

observed in global myelination (*Figure 5—figure supplement 1a,b*) or CC1+ mature oligodendrocyte density (*Figure 5—figure supplement 1c,d*).

Systematic analysis of individual PV+ interneuron axonal segments in PV::UBE3A and PV::TSC1 cells confirmed a similarly strong co-variation between axon morphology and segmental myelination as observed in PV::WT mice (*Figure 5o–s*). ROC analysis of PV::UBE3A cells yielded bivariate thresholds of axonal diameter >332 nm and interbranch distance >14.1 μm (sensitivity = 100%, specificity = 97.5%; AUC = 0.99) (*Figure 5p,q*). Analogously, ROC analysis of the PV::TSC1 cells yielded bivariate thresholds of axonal diameter >378 nm and interbranch distance >18.6 μm (sensitivity = 100%, specificity = 94.6%; AUC = 0.99) (*Figure 5r,s*). Similar to PV::WT, the bivariate model for PV::TSC1 significantly improved the prediction accuracy of segmental myelination compared to the univariate models (axonal diameter: p=0.015; interbranch distance: p=0.027). For PV::UBE3A, the bivariate model was a significant improvement over the univariate model based on interbranch

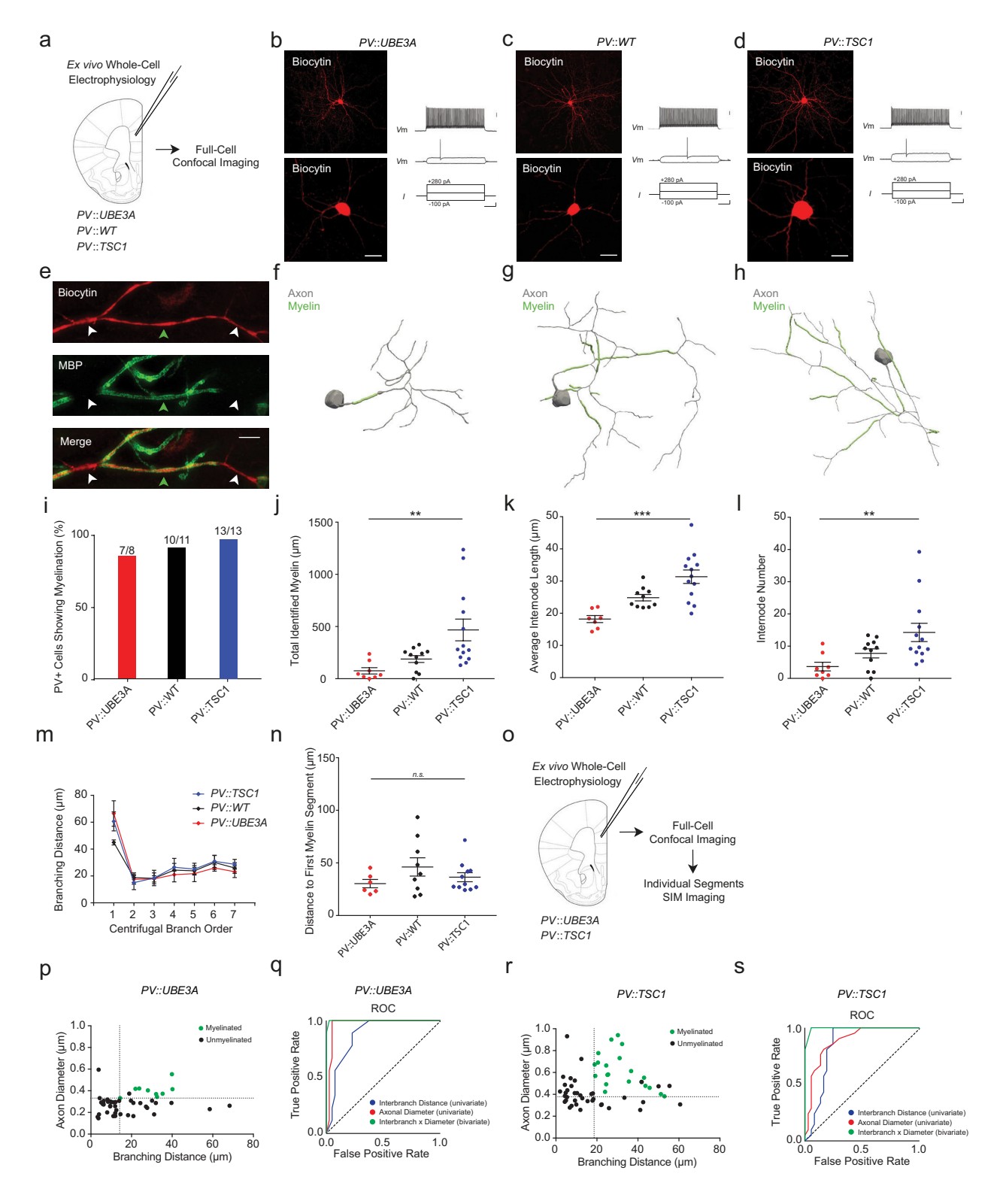

**Figure 5.** PV::UBE3A and PV::TSC1 mice exhibit bi-directional alterations in PV+ interneuron axonal myelination. (a) Experimental approach. Biocytin-filled fast-spiking PV+ interneurons from each genotype were first analyzed using confocal imaging. (b–d) Maximum projection image of a representative biocytin-filled PV+ cell (red, top), a close-up of a biocytin-filled somata (red, bottom), with a corresponding fast-spiking action potential train for PV::UBE3A (b), PV::WT (c), and PV::TSC1 (d). Scale bars are 50 μm (left) and 20 mV, 100 pA and 100 ms from top to bottom (right).
*Figure 5 continued on next page*

*Figure 5 continued*

(e) Representative SIM *z*-stack projection of a biocytin-filled PV+ interneuron axon centered over a 4th order branch (red), demonstrating myelinated (MBP, green; green arrowhead) and unmyelinated segments (white arrowhead). Scale bar, 5 µm. (f–h) Proximal axon reconstructions (grey) including myelinated segments (green) of representative cells from PV::UBE3A (f), PV::WT (g), and PV::TSC1 (h) mice. (i) Nearly all PV+ cells exhibit axonal myelination, independent of genotype. (j–l) PV+ cell-specific genetic manipulations bi-directionally alter myelin content (j), internode length (k) and number of internodes per cell (l). (m) The distribution of interbranch distance remains similar across genotypes (p=0.575, repeated measures ANOVA group x branch order interaction). (n) Distance from the soma to the onset of myelination was unaffected by PV+ cell-specific deletion of *Ube3a* or *Tsc1*. p=0.589, one-way ANOVA. (o) Experimental approach. Biocytin-filled fast-spiking PV+ interneurons from PV::UBE3A and PV::TSC1 mice were analyzed using both confocal and SIM imaging. (p) The joint combination of axonal diameter and interbranch point distance is highly predictive of PV::UBE3A cell segmental myelination. Critical thresholds (dotted lines) for interbranch distance and axonal diameter were 14.1 µm and 332 nm, respectively. *n* = 49 segments/3 cells. Myelinated segments, green circles. Unmyelinated segments, black circles. (q) Receiver-operator characteristic (ROC) curve for PV::UBE3A cells. ROC curves of segmental myelination, comparing univariate models of interbranch distance (blue) and axonal diameter (red), and their joint bivariate combination (green). Diagonal dotted line indicates the non-discrimination reference boundary. (r) The joint combination of axonal diameter and interbranch point distance is highly predictive of PV::TSC1 cell segmental myelination. Critical thresholds (dotted lines) for interbranch distance and axonal diameter were 18.6 µm and 378 nm, respectively. *n* = 58 segments/3 cells. Myelinated segments, green circles. Unmyelinated segments, black circles. (s) ROC curves for PV::TSC1 cells. ROC curves of segmental myelination, comparing univariate models of interbranch distance (blue) and axonal diameter (red), and their joint bivariate combination (green). Diagonal dotted line indicates the non-discrimination reference boundary. ***p<0.001, **p<0.01, *p<0.05, n.s. non-significant. One-way ANOVA in (j), (k), (l) and (n). Repeated measures ANOVA in (m). Black bars represent mean ± s.e.m. Abbreviations: I, input current. ROC, receiver-operator characteristic. Vm, membrane voltage.

The online version of this article includes the following source data and figure supplement(s) for figure 5:

**Source data 1.** Total recovered myelination length (j), internode length (k), internode number (l), branch order (m), myelin onset distance (n), as well as bivariate interbranch distance / axonal diameter values for myelinated and unmyelinated segments of PV::UBE3A (p) and PV::TSC1 (q) cells.

**Figure supplement 1.** Intact global myelination and mature oligodendrocyte density in mPFC of mice with PV cell-specific mutations of *Ube3a* or *Tsc1*.

distance (p=0.004), but statistically similar to the univariate model for axonal diameter (p=0.500). Together, these findings demonstrate the robustness of the joint combination of interbranch distance and axonal diameter for accurately predicting the topography of PV+ interneuron myelination in vivo, across a wide range of axonal morphologies.

## SOM+ and PV+ interneurons adhere to similar myelination rules

Regular spiking, SOM+ interneurons (*Urban-Ciecko and Barth, 2016*) have relatively few myelinated internodes along their axons in mPFC (*Stedehouder et al., 2017*) and contribute minimally to total neocortical myelination content (*Micheva et al., 2016*). Is this sparse myelination also predicted by their axonal morphology, as we observed for PV+ interneurons? Do SOM+ interneuron axons share the thinner shaft diameters or more closely spaced branch points of unmyelinated segments of PV+ axons?

To examine this possibility, we performed whole-cell recordings and intracellular biocytin-labeling of SOM+ interneurons in mPFC using *Sst*::cre (*Taniguchi et al., 2011*),Ai14 (*Madisen et al., 2012*) mice (*n* = 10; *Figure 6a–d*; *Figure 1—figure supplement 3*). The identity of filled cells was further confirmed as SOM+ interneurons based upon their characteristic electrophysiological and morphological features (*Urban-Ciecko and Barth, 2016*) (*Figure 6c,d*; *Supplementary file 2*; *Figure 7—figure supplement 1e*). Among the 10 reconstructed SOM+ axons, we found only a single myelinated internode (*Figure 6e–h*). Axonal shaft diameter as quantified using SIM averaged 0.303 ± 0.015 µm (range 0.222–0.561 µm; *Figure 6f*, *Figure 7—figure supplement 1f*) and decreased with increasing branch order (*Figure 6g*). Notably, the vast majority of unmyelinated SOM+ axonal segments had smaller diameters and/or more closely spaced branch points than the threshold values identified among myelinated segments in PV+ interneurons (*Figure 6h* inset; dashed line).

Since we identified only a single myelinated segment, we could not reliably determine critical thresholds for axonal diameter and interbranch distance thresholds among wild-type SOM+ interneurons. Notably, however, this single myelinated segment had an axonal diameter of 561 nm and interbranch distance of 16.8 µm, which exceeded the morphometric thresholds we identified for wildtype PV+ interneurons. Similarly, the vast majority of unmyelinated SOM+ interneuron axon segments fell below morphometric thresholds for axonal diameter and/or interbranch distance identified for PV::WT cells (79 of 88 segments; 89.8%) (*Figure 6h*). These data indicate that the myelination rules derived from the analysis of PV+ cells may be generalizable to other neocortical interneurons, including SOM+ cells.

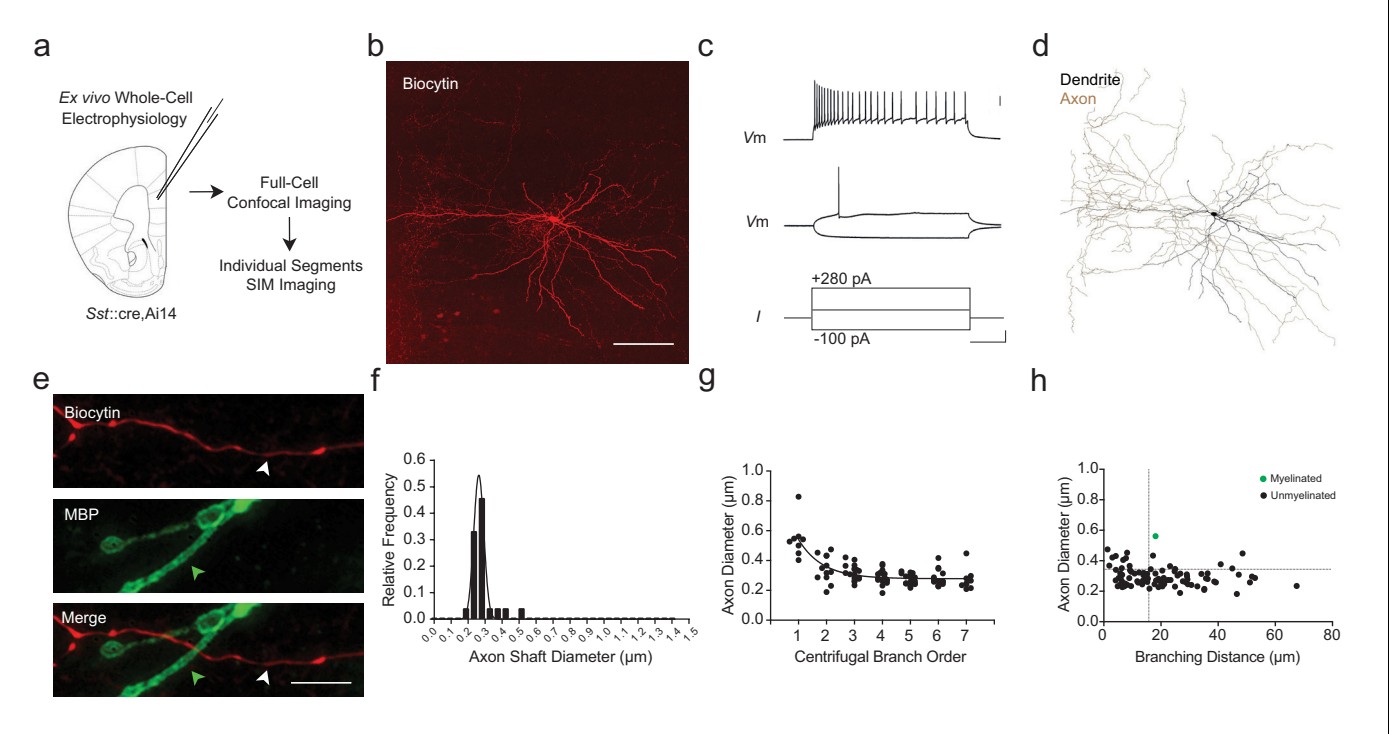

**Figure 6.** mPFC SOM+ interneurons have subthreshold axonal morphology and are correspondingly unmyelinated. (**a**) Experimental approach. Biocytin-filled regular-spiking SOM+ interneurons from mPFC were analyzed using both confocal imaging and SIM imaging. See also *Figure 1—figure supplements 1–3*. (**b**) Maximum projection confocal image of a representative biocytin-filled SOM+ interneuron (red). Scale bar, 50 μm. (**c**) SOM+ interneuron recording demonstrating a low threshold for AP initiation, spike frequency adaptation, and AP amplitude attenuation. Scale bars are 20 mV, 100 pA and 100 ms from top to bottom. (**d**) Neurolucida reconstruction of the SOM+ cell depicted in (**b**). Soma and dendrite in black, axon in brown. Note the tortuous axonal arbor and dendrites. (**e**) mPFC SOM+ interneurons are rarely myelinated. Representative confocal image of a SOM+ interneuron axon (red), centered over a 3rd branch order segment (white arrowhead) without myelination (MBP, green; green arrowhead). Scale bar, 10 μm. (**f**) Frequency histogram of SOM+ interneuron axon shaft diameter, fitted with a Gaussian curve. *n* = 88 axonal segments/6 cells. (**g**) Axon shaft diameter decreases monotonically with increasing centrifugal branch order. *n* = 88 segments/6 cells. p<0.001, one-way ANOVA. (**h**) Distribution of axonal segment diameter and interbranch distances for myelinated (green circles) and unmyelinated (black circles) segments. *n* = 88 segments/6 cells. Dotted lines indicate the bivariate thresholds derived from *PV::WT* interneurons. Abbreviations: I, input current. Vm, membrane voltage.
The online version of this article includes the following source data for figure 6:

**Source data 1.** Diameter measurements for axonal segments (f), branch order (g), and bivariate interbranch distance / axonal diameter values for myelinated and unmyelinated segments (h) of SOM::WT cells.

## Genetic manipulation of SOM+ axon morphology induces de novo myelination

Factors other than axon morphology could explain why SOM+ interneurons are largely unmyelinated, such as expression of active inhibitors of myelination (*Redmond et al., 2016*). We therefore asked whether we could induce de novo myelination of SOM+ cells by altering their axonal morphology. Using an analogous approach to that we used for PV+ cells, we deleted *Tsc1* specifically in SOM+ cells by crossing *Sst*::cre and floxed *Tsc1^{f/f}* mice. SOM+ interneurons identified by *post hoc* immunolabeling had ~65% larger soma size in mPFC compared to WT mice (*Figure 7a,b*; *Figure 7— figure supplement 1*). Moreover, cre-dependent viral labeling of SOM+ cells followed by SIM imaging showed increased axon calibers for SOM::TSC1 cells (1st order branches) compared to WT mice (*Figure 7c,d*). SOM+ cell density was unchanged in the mPFC of SOM::TSC1 mice (*Figure 7—figure supplement 2a,b*).

We next performed whole-cell electrophysiological recordings in mPFC SOM+ cells combined with biocytin-filling and *post hoc* MBP immunofluorescence to examine the influence of the enlarged morphology on axonal myelination (*Figure 7e,f*; *Figure 1—figure supplement 3*; *Figure 7—figure*

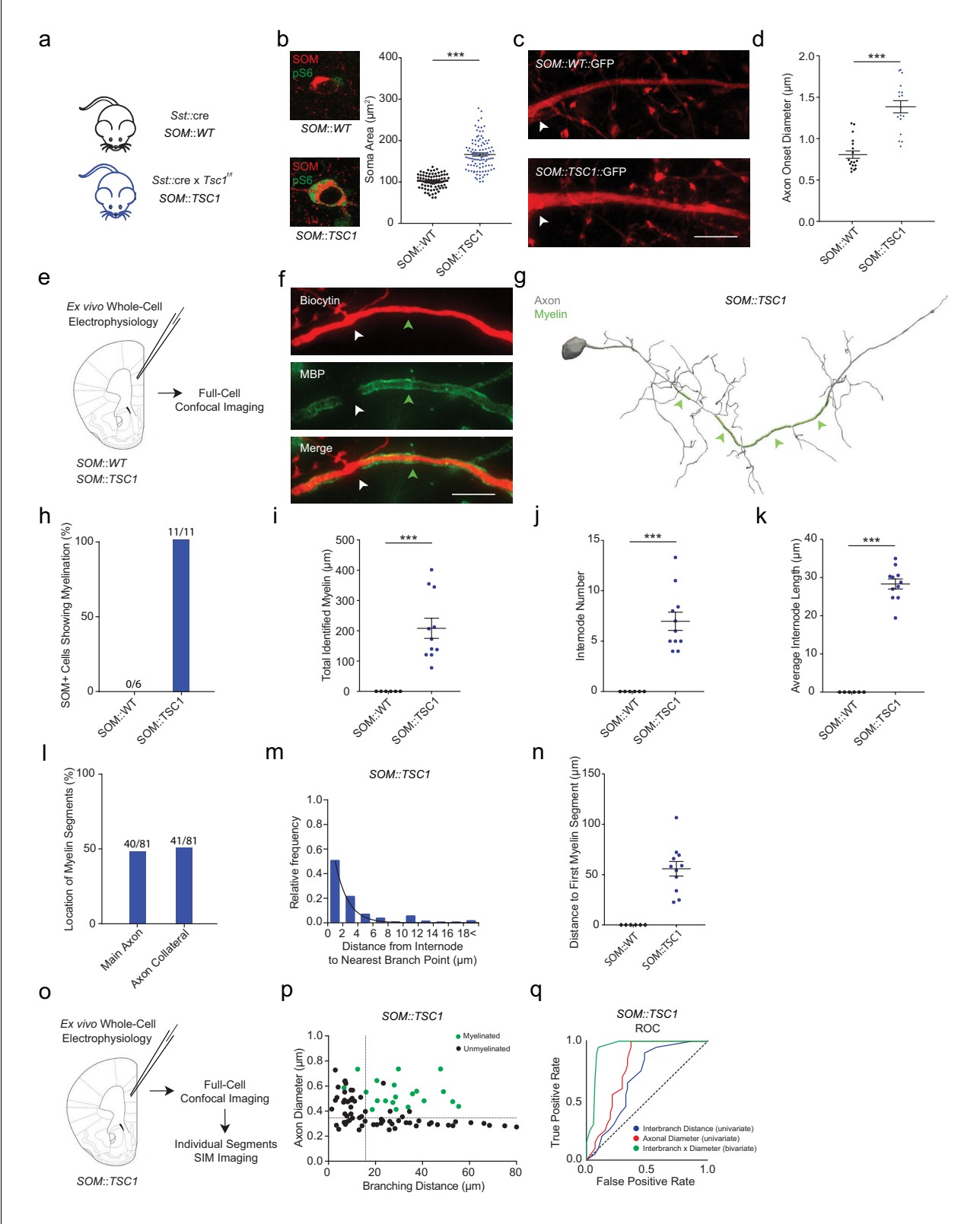

**Figure 7.** SOM::TSC1 cells are frequently myelinated. (**a**) Overview of mouse breeding scheme. (**b**) SOM::TSC1 cells have enlarged somata. Left: Representative confocal images of SOM+ cells (red) confirm the increased expression of pS6 (green) in SOM::TSC1 cells. Higher magnification images in *Figure 7—figure supplement 1*. Right: SOM+ interneuron maximum projection soma area from mPFC layers II-V. SOM::WT: $101.8 \pm 1.9\ \mu m^2$, $n = 81$ cells; SOM::TSC1: $166.4 \pm 3.7\ \mu m^2$, $n = 103$ cells; $n = 3$ mice per group. ***p<0.001. Unpaired two-tailed Mann-Whitney $U$-test. (**c**) Representative SIM *z-Figure 7 continued on next page*

*Figure 7 continued*

stack projection images of GFP-labelled SOM+ interneuron axons (red). SOM::WT (top), SOM::TSC1 (bottom). Scale bar, 3 μm. (d). SOM::TSC1 cells have an increased axonal diameter. SOM::WT: 0.799 ± 0.049 μm, n = 18 cells; SOM::TSC1: 1.407 ± 0.076 μm, n = 17 cells; n = 3 mice per group. ***p<0.001. Unpaired two-tailed Mann-Whitney *U*-test. (e) Experimental approach. Biocytin-filled regular-spiking SOM+ interneurons from mPFC of SOM::WT and SOM::TSC1 mice were analyzed using confocal imaging (f) Representative confocal *z*-stack projection image of a biocytin-filled SOM::TSC1 interneuron axon (red) and myelination (MBP), centered over a 7th branch order segment. Unmyelinated branch point indicated by white arrowhead. Scale bar = 10 μm. (g) Neurolucida reconstruction of an mPFC SOM::TSC1 cell. The axon (grey) shows multiple myelinated internodes (green) along the main branch. Note the frequent thin, tortuous, and unmyelinated axon collaterals. (h) In contrast to SOM::WT cells, all 11 reconstructed SOM::TSC1 cells were myelinated. (i–k) SOM::TSC1 cells exhibit a robust (i) total myelin (208.30 ± 33.14 μm), (j) number of internodes (6.97 ± 0.90 μm), and (k) internode length (28.33 ± 1.31 μm). ***p<0.001, Unpaired Student's two-tailed *t*-test. SOM::WT: n = 6; SOM::TSC1: n = 11. (l) Myelin segments occur with similar frequency on main SOM::TSC1 axon branches and axon collaterals (p=0.99, Fisher's Exact Test). (m) Frequency histogram of nearest neighbor distance from internodes to branch points. n = 38 segments/4 cells. (n) Distance from the soma or originating dendrite to the onset of myelination was measurable only for SOM::TSC1 cells due to the very infrequent myelination of SOM::WT cells. (o) Experimental approach. Biocytin-filled regular-spiking SOM+ interneurons from SOM::TSC1 mice were analyzed using confocal and SIM imaging. (p) The joint combination of axonal diameter and interbranch point distance is highly predictive of PV::TSC1 cell segmental myelination. Critical thresholds (dotted lines) for interbranch distance and axonal diameter were 11.8 μm and 406 nm, respectively. n = 86 segments/5 cells. Myelinated segments, green circles. Unmyelinated segments, black circles. (q) ROC curves for SOM::TSC1 cells. ROC curves of segmental myelination, comparing univariate models of interbranch distance (blue) and axonal diameter (red), and their joint bivariate combination (green). Diagonal dotted line indicates the non-discrimination reference boundary. Unpaired Student's two-tailed *t*-test in (i), (j) and (k). Unpaired two-tailed Mann-Whitney *U*-test in (b) and (d) owing to non-normality. Black bars represent mean ± s.e.m. Abbreviations: ROC, receiver-operator characteristic.

The online version of this article includes the following source data and figure supplement(s) for figure 7:

**Source data 1.** Soma area (b), axon onset diameter (d), total recovered myelination length (i), internode number (j), internode length (k), myelin onset distance (n), as well as bivariate interbranch distance / axonal diameter values for myelinated and unmyelinated segments of SOM::TSC1 (p) cells.

**Figure supplement 1.** SOM::TSC1 mice exhibit SOM-specific deletions of Tsc1.

**Figure supplement 2.** SOM::TSC1 mice have a normal SOM+ cell density in mPFC.

**Figure supplement 3.** Electrophysiological properties of SOM::TSC1 and SOM::WT cells.

---

supplement 1). Recorded SOM::TSC1 cells exhibited reduced input resistance and intrinsic excitability, with no changes in single action potential characteristics (*Figure 7—figure supplement 3*).

Confocal microscopy followed by axonal reconstruction showed that whereas SOM::WT cells were rarely myelinated (0 out of 6; 0%), which is in line with the *Sst*::cre,Ai14 cells, myelination of SOM::TSC1 cells was highly frequent (11 out of 11; 100%; *Figure 7f–h*). Moreover, SOM::TSC1 cells showed corresponding increases in total length of myelination (*Figure 7i*), internode length (*Figure 7k*), and number of internodes (*Figure 7j*). Myelin onset appeared at 55.8 ± 7.2 μm from the soma (*Figure 7n*), typically initiating between the 2nd and 6th branch order. No myelin was identified on more distal axonal segments (branch order ≥10). Myelination was found equally on the primary axon and on axon collateral branches (*Figure 7l*). Furthermore, SOM::TSC1 myelination was constrained by axonal branch points, in that 87% of internodes began or ended within 5 μm of a branch point (*Figure 7m*).

Analyses of individual axonal segments using SIM along SOM::TSC1 cells revealed a similar relationship between myelination and the joint combination of axonal diameter and interbranch distance (*Figure 7o–q*) as found in wildtype PV+ cells (*Figure 2f*, *Figure 5n,o*). ROC analysis of SOM::TSC1 cells yielded thresholds of axonal diameter >406 nm and interbranch distance >11.8 μm (sensitivity = 0.95, specificity = 0.90; AUC = 0.94) (*Figure 7p*). The bivariate model exhibited a significantly improved the prediction accuracy for segmental myelination compared to the univariate models (axonal diameter: p<0.001, interbranch distance: p<0.001). These findings suggest that axonal morphology, and in particular the combination of axonal caliber and interbranch distance, governs the local segmental myelination of PV+ and SOM+ neocortical interneurons.

Neocortical SOM+ interneurons are morphologically and electrophysiologically heterogeneous (*Jiang et al., 2015*). Therefore, to further examine the extent of SOM::TSC1 myelination across a wider population of cells, we employed SOM-specific sparse viral transduction using cre-dependent GFP expression in adult mPFC, followed by MBP immunofluorescence (*Figure 8a*). Although this method precludes electrophysiological confirmation of adapting spiking patterns and detailed axonal reconstructions, it provides a higher-throughput examination of mPFC SOM+ interneuron myelination. We examined axons originating from mPFC layer II-V GFP+ interneuron somata from SOM::WT and SOM::TSC1 mice for colocalization with MBP (*Figure 8b*). Of 26 SOM::WT+ cells examined,

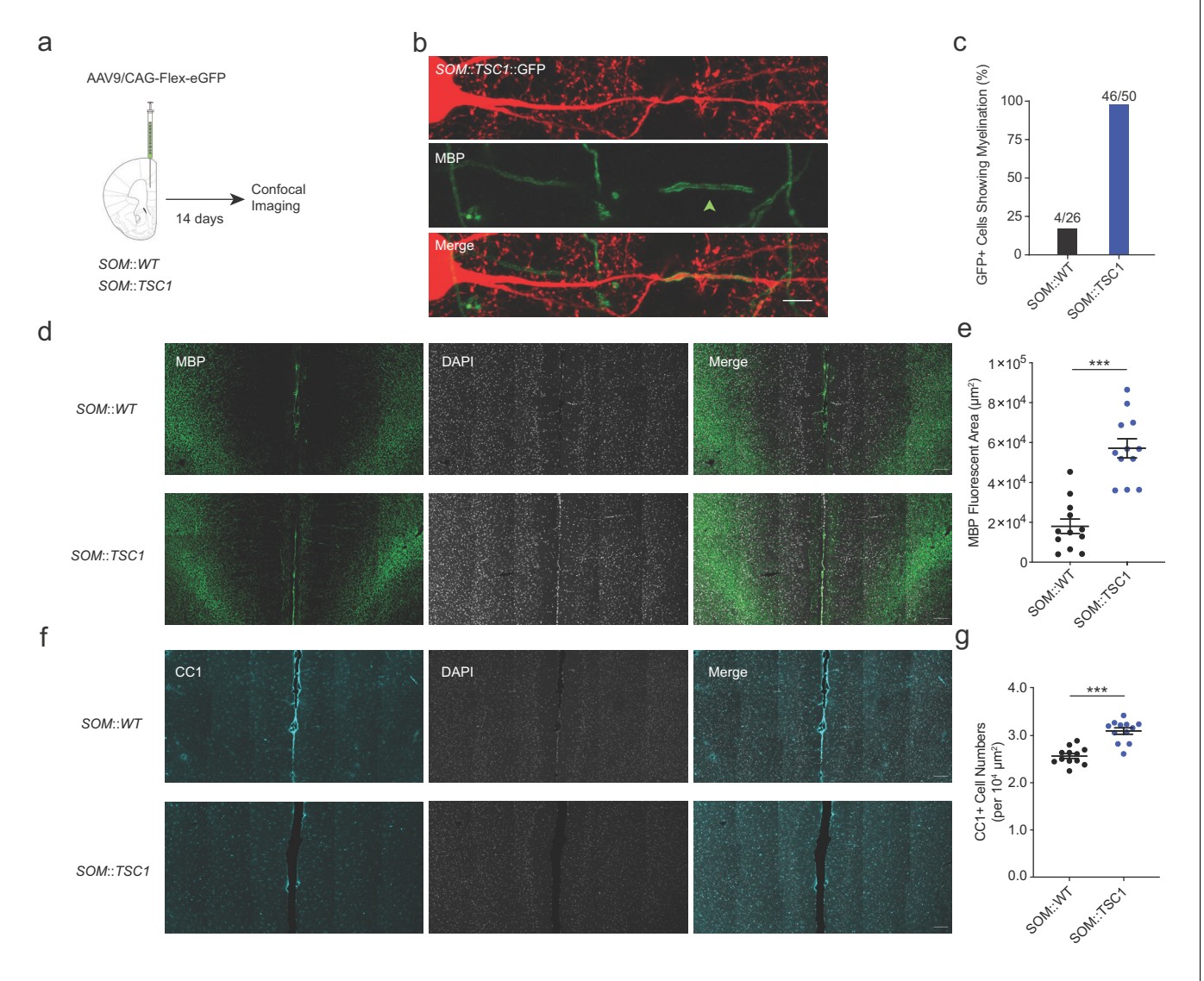

**Figure 8.** Extensive axonal myelination of SOM::TSC1 cells. (a) Experimental flowchart. Adult SOM::WT and SOM::TSC1 mice were injected with low-titer AAV9 cre-dependent GFP reporter virus in mPFC, and sacrificed 14 days later for MBP immunofluorescence labeling and confocal microscopy. (b) Maximum projection confocal image of a GFP-labelled SOM::TSC1+ interneuron showing circumferential MBP expression (green). Scale bar, 5 μm. (c) A high proportion of virally-labelled SOM::TSC1 cells exhibited myelination (92.0%, 46 of 50 cells), in contrast to SOM::WT cells (15.4%, 4 of 26 cells) (p<0.001, Fisher's Exact Test). (d) Representative low-magnification confocal image from mPFC showing the robust increase of myelination (MBP, green) in SOM::TSC1 compared to SOM::WT mice. DAPI in white. Scale bar, 100 μm. (e) Quantification of MBP+ area in mPFC of SOM::TSC1 mice (5.7 × $10^4$ ± 0.48 x $10^4$ μm$^2$) compared to SOM::WT mice (1.8 × $10^4$ ± 0.36 x $10^4$ μm$^2$). p<0.001, Unpaired Student's two-tailed *t*-test. (f) Confocal microscopy image showing immunofluorescence of CC1 (cyan) and DAPI (white) in adult mPFC of SOM::WT and SOM::TSC1 mice. Scale bar, 100 μm. (g) CC1+ cell density in adult mPFC of SOM::TSC1 and SOM::WT mice. p<0.001, Unpaired Student's two-tailed *t*-test. Black bars indicate mean ± s.e.m.
The online version of this article includes the following source data for figure 8:

**Source data 1.** MBP+ area (e) and CC1+ cell counts (g) in SOM::WT and SOM::TSC1 cells.

4 (15%) were myelinated (*Figure 8c*). Conversely, of 50 SOM::TSC1+ cells examined, 46 (92%) were myelinated (p<0.001; *Figure 8c*). Consistent with the high proportion of SOM+ cell myelination, the mPFC of SOM::TSC1 exhibited a marked increase of global myelination (*Figure 8d,e*) and a higher density of CC1+ mature oligodendrocytes (*Figure 8f,g*).

Together, these data suggest that axonal morphology is necessary and sufficient to govern neo-cortical interneuron myelination. In particular, we propose that the combination of interbranch

distance and axonal diameter together determine the topographical distribution of internodes along the axons of neocortical interneurons.

## Human neocortical interneurons adhere to similar myelination rules

We next examined whether the above findings in mice also extend to human neocortex. Using ex vivo resected tissue from patients undergoing tumor surgery, we performed whole-cell electrophysiological recordings (*Supplementary file 3*) and intracellular biocytin labeling with *post hoc* MBP immunofluorescence of fast-spiking interneurons classified on the basis of morphology and electrophysiology (*Figure 9a–e*; 4 cells from four donors; see Materials and methods).

Reconstructed human fast-spiking interneurons exhibited a similar total length of myelination (432.0 ± 137.9 µm), average internode length (44.7 ± 4.9 µm), and number of internodes (9.7 ± 1.9 internodes) as previously described (*Stedehouder et al., 2017*). Myelin onset appeared at 36.2 ± 13.3 µm from the soma, initiating between the 1st (2 out of 4 cells; 50%) and 3rd branch order (2 out of 4 cells; 50%). Analogous to mouse interneurons, no myelin was identified on more distal axonal segments (branch order ≥10; *Figure 9f*).

Analyses of individual axonal segments using SIM along human fast-spiking interneurons revealed a similar relationship between myelination and the joint combination of axonal diameter and interbranch distance (*Figure 9h–j*) as found in mouse PV+ cells (*Figure 2f*). ROC analysis of human interneurons yielded thresholds of axonal diameter >328 nm and interbranch distance >13.7 µm (sensitivity = 92.5%, specificity = 89.3%; AUC = 0.96; *Figure 9k*). The accuracy of the bivariate model (90.6%, 87 of 96 segments) was significantly higher than interbranch distance (66.7%, p<0.001) but non-significantly different from axonal diameter (83.3%, p=0.198). Together, these findings suggest that mouse and human neocortical interneurons follow similar morphological rules guiding the topography of axonal myelination.

## Discussion

Fast-spiking, PV-positive interneurons are frequently myelinated in the cerebral cortex, and their myelination forms a considerable proportion of cortical myelin (*Micheva et al., 2016*; *Stedehouder et al., 2017*; *Stedehouder and Kushner, 2017*). PV+ myelination exhibits a proximally-biased topography consisting of short internodes interspersed by branch points, whereas more distal axonal segments decorated with frequent *en passant* boutons remain unmyelinated (*Stedehouder et al., 2017*; *Stedehouder et al., 2018*). Conversely, other interneuron subclasses, such as irregularly spiking SOM-positive interneurons, are sparsely myelinated and contribute minimally to the total content of neocortical myelin (*Stedehouder et al., 2017*). However, it has remained unknown why PV+ interneurons are preferentially myelinated compared to other neocortical interneuron subtypes. Here, we have provided evidence suggesting that axonal morphology is a strong determinant of the topography of myelination along individual axons.

Using single-cell axonal reconstructions, we revealed a high co-variation between segmental myelination and the joint combination of interbranch distance (~14 µm) and axonal diameter (~330 nm) thresholds. These parameters were remarkably similar across two interneuron subtypes (PV *vs.* SOM) and independent of genetic manipulation of their morphologies. Moreover, our results appear to provide an explanation for the proximally-biased topography of PV+ interneuron myelination, in which internodes are consistently present at the first axonal branch order with a declining probability at increasing branch orders (*Stedehouder et al., 2017*; *Stedehouder et al., 2018*). Given that axonal shaft diameter decreases with increasing branch order (see *Figure 1h*, but also *Hu and Jonas, 2014*), distal axonal segments would therefore remain unmyelinated by virtue of their thinner axon shafts, despite often retaining supra-threshold interbranch distances. Differences in axon morphology presumably also account, at least in part, for the more robust myelination of neocortical PV+ interneurons relative to other interneuron subtypes (*Micheva et al., 2016*; *Stedehouder et al., 2017*). However, it remains unknown whether the myelination rules revealed here for interneurons extends to neocortical pyramidal cells, which also exhibit interspersed unmyelinated segments (*Tomassy et al., 2014*). Prior studies using serial EM have identified a ~ 300 nm threshold of axon diameter for both GABAergic as well as non-GABAergic axons, suggesting the morphological parameters could well extend to excitatory axons (*Micheva et al., 2016*).

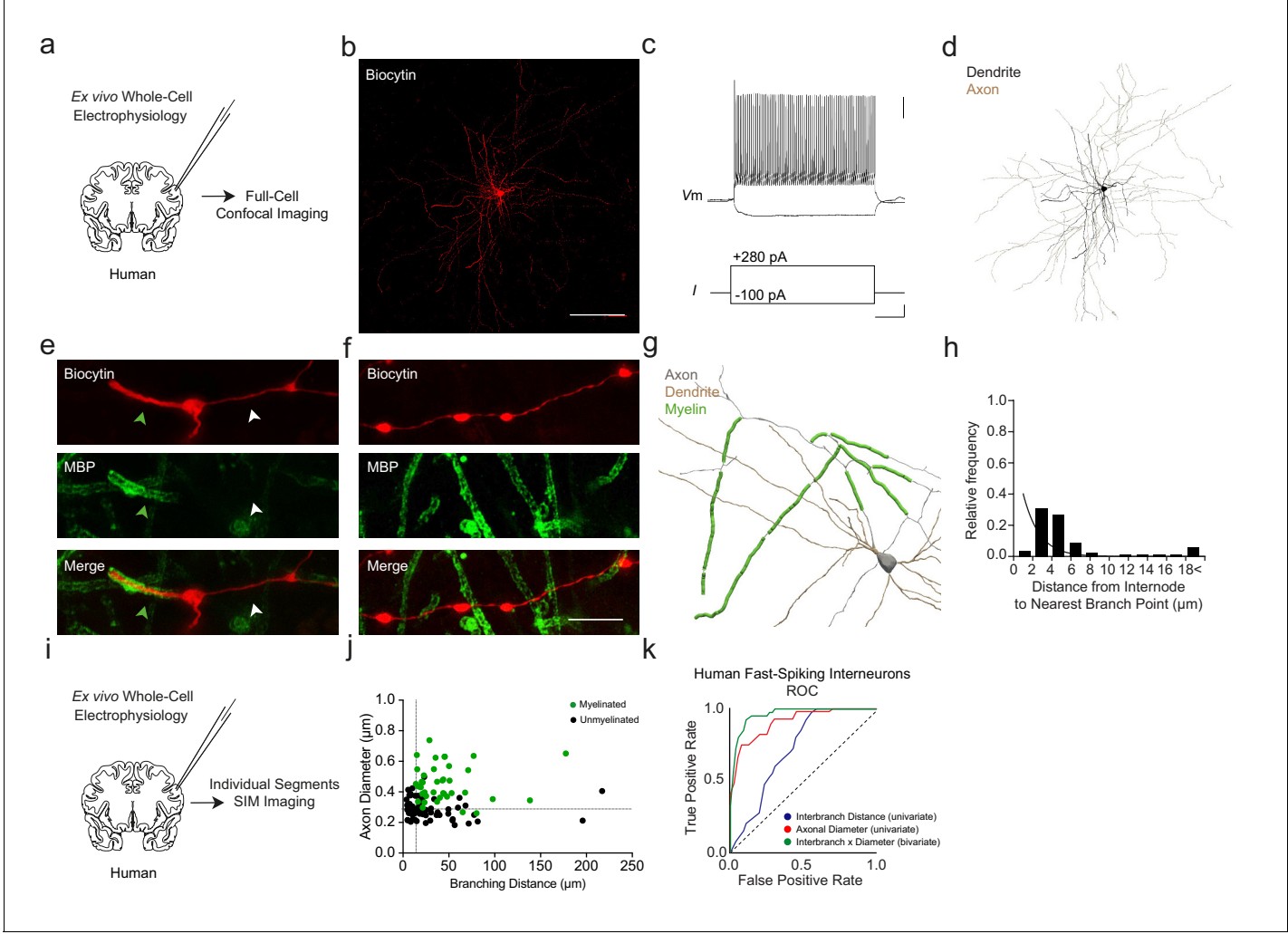

**Figure 9.** Myelination thresholds extend to fast-spiking interneurons in human cortex. (a) Experimental approach. Biocytin-filled interneurons from human ex vivo resected cortical tissue were analyzed using confocal imaging. (b) Maximum projection confocal image of a representative biocytin-filled human fast-spiking interneuron (red). Scale bar, 50 µm. (c) Current clamp recording of evoked action potentials of human fast-spiking interneuron. Scale bars are 20 mV, 100 pA and 100 ms from top to bottom (right). (d) Full reconstruction of a human fast-spiking interneuron. Soma and dendrites in black, axon in brown. (e) Representative SIM z-stack projection image of a PV+ cell (biocytin, red), centered over a myelinated (MBP; green) 1st order axonal segment. Note that the myelinated axonal segment (green arrowhead) has a larger diameter than the unmyelinated axon segment (white arrowhead). (f) Representative SIM z-stack projection featuring an unmyelinated segment of axon (red) centered over the 3rd branch order. Scale bar 10 µm for (e) and (f). (g) Neurolucida reconstruction of a human fast-spiking interneuron axon. Axon in grey, myelinated segments in green. Since axon originated from proximal dendrite, dendrite is also depicted in brown. Note the proximal onset of myelin, consisting of short internodes interspersed by branch points. (h) Frequency histogram of nearest neighbor distance from internodes to branch points. n = 55 segments/3 cells. (i) Experimental approach. Biocytin-filled interneurons from human ex vivo resected cortical tissue were analyzed using SIM imaging. (j) The joint combination of axonal diameter and interbranch point distance is highly predictive of human fast-spiking interneuron segmental myelination. Critical thresholds (dotted lines) for interbranch distance and axonal diameter were 13.7 µm and 328 nm, respectively. n = 96 segments/4 cells. Myelinated segments, green circles. Unmyelinated segments, black circles. (k) ROC curves for human fast-spiking interneurons. ROC curves of segmental myelination, comparing univariate models of interbranch distance (blue) and axonal diameter (red), and their joint bivariate combination (green). Diagonal dotted line indicates the non-discrimination reference boundary. Abbreviations: ROC, receiver-operator characteristic. I, input current. Vm, membrane voltage.
The online version of this article includes the following source data for figure 9:

**Source data 1.** Morphological measures in human fast-spiking neocortical interneurons: internode-to-branch point (h), and bivariate interbranch distance / axonal diameter values for myelinated and unmyelinated segments (j).

We find here that an uninterrupted interbranch distance of ~14 µm is a strict requirement for segmental myelination in both PV+ and SOM+ interneurons. This threshold approximates the length of the smallest myelinated internodes found on neocortical axons of PV+ interneurons (*Stedehouder et al., 2017*) and pyramidal neurons (*Tomassy et al., 2014*), and more generally across unspecified cell types in the cerebral cortex (*Chong et al., 2012*). Notably, pre-myelination nodal clustering – the clustering of voltage-gated sodium channels which later form nodes of Ranvier – exists in similar intervals (*Freeman et al., 2015*). One possibility is that ~14 µm is the minimum biophysical length compatible with oligodendrocyte ensheathment.

The functional consequence of cortical interneuron myelination is currently unknown. Fast-spiking PV+ interneurons have demanding metabolic requirements (*Kann et al., 2014*). As myelination has been established to function critically in providing axonal metabolic support (*Lee et al., 2012b*; *Fünfschilling et al., 2012*; *Saab et al., 2016*), myelination of PV+ interneurons could function by helping to optimize axonal energy utilization. Moreover, PV+ interneurons coordinate fast synchronizing network activity in the gamma frequency range (*Hu et al., 2014*), for which segmental interneuron myelination could enhance synchronized inhibition through local modulation of action potential conduction velocity. This is particularly interesting given recent findings that distance-dependent inhibition functions in regulating gamma synchrony (*Strüber et al., 2015*).

Only a small fraction of neocortical myelin localizes along the axons of irregular-spiking SOM+ interneurons, even though these cells represent ~30% of interneurons (*Markram et al., 2004*). Accordingly, we found that the limited myelination of SOM+ interneurons was highly consistent with their generally thinner axon diameter and shorter interbranch distances compared to PV+ cells. Enlargement of SOM+ interneurons by cell-type restricted deletion of *Tsc1* led to a dramatic increase in the frequency and extent of axonal myelination. Notably, this de novo myelination of SOM::TSC1 cells was also accurately predicted by the same joint axonal diameter and interbranch distance parameters as identified in PV+ interneurons. However, compared to the PV::TSC1 cell manipulation, SOM::TSC1 mice exhibited a more robust increase of global myelination in the mPFC, which was accompanied by an increased number of CC1+ mature oligodendrocytes. One possibility for this finding is that PV$^+$ cell morphology is already largely optimized for supra-threshold morphology permissive of myelination, in contrast to SOM$^+$ cells which are predominantly unmyelinated under normal conditions and therefore exhibit a greater magnitude increase of myelination when their morphology is enlarged by deletion of *Tsc1*. A second non-mutually exclusive possibility is that SOM::TSC1 deletion results in additional non-cell autonomous effects that enhance global myelination and recruitment of mature oligodendrocytes.

An important outstanding question is whether the axonal morphology rules of PV+ and SST+ interneurons extend to neocortical glutamatergic pyramidal cells (*Timmler and Simons, 2019*). Some evidence exists to suggest that analogous morphological thresholds might also extend to non-GABAergic cells in the neocortex (*Micheva et al., 2016*; *Micheva et al., 2018*). For example, the more frequently myelinated first-order axonal segment of PV$^+$ GABAergic axons has a ~ 2.5 fold larger caliber than glutamatergic pyramidal neurons (*Schmidt et al., 2017*). However, the highly discontinuous topography of internodes along glutamatergic pyramidal cells also suggests a major influence of factors beyond morphology (*Tomassy et al., 2014*). One possibility is that myelination of pyramidal neurons involves a combination of attractive and/or repulsive molecular cues, such as neuregulin-ErbB signaling (*Lundgaard et al., 2013*) or glutamatergic signaling (*Gautier et al., 2015*; *Habermacher et al., 2019*; *Kougioumtzidou et al., 2017*; *Chen et al., 2018*; *Berret et al., 2017*), in addition to axonal morphological thresholds (*Makinodan et al., 2012*). Furthermore, a recent study observed that a subset of neocortical oligodendrocytes exclusively ensheath GABAergic interneurons, raising the possibility that distinct subtypes of oligodendrocytes might have neuronal cell type-specific rules governing axonal myelination (*Zonouzi et al., 2019*).

Myelination of neocortical glutamatergic pyramidal neurons is modulated in vivo by experience, as well as direct manipulation of neuronal activity (*Fields, 2015*). Activity-dependent myelination also extends to neocortical PV+ interneurons, a finding associated with concordant alterations in axonal morphology (*Stedehouder et al., 2018*). When considered together with the current findings, it is therefore possible that neuronal activity-dependent myelination (*Gibson et al., 2014*; *Mitew et al., 2018*) in the cerebral cortex might be mediated by axonal morphological plasticity.

The current studies were focused primarily on locally-projecting GABAergic interneurons in pre-frontal cortex. However, several studies have identified long-range GABAergic projecting cells including, but not limited to, PV+ and SOM+ subclasses (*Caputi et al., 2013*; *Melzer et al., 2017*; *Rock et al., 2016*; *Lee et al., 2014*). The high proportion of cerebral cortex PV+ cells exhibiting axonal myelination (*Stedehouder et al., 2017*) makes it likely that long-range PV+ cells are also myelinated. However, due to technical limitations of slicing and intracellular biocytin labeling, it was not possible to definitively identify and reconstruct long-range GABAergic axons. Thus, the question remains open whether long-range GABAergic axons are also frequently myelinated and whether axonal morphology is similarly predictive of their myelination. Moreover, a related issue regards whether the present findings acquired in grey matter also extend to white matter, especially given regional differences among oligodendrocyte lineage cells (*Dimou and Simons, 2017*; *Spitzer et al., 2019*).

Myelination of GABAergic interneurons has been observed in multiple mammalian species (*Stedehouder and Kushner, 2017*). Notably, human neocortical fast-spiking PV+ interneurons appear to have more extensive total myelination per cell than is observed in mice (*Stedehouder et al., 2017*; *Micheva et al., 2018*). This raises the possibility that neocortical interneuron myelination might have a crucial influence on higher cognitive function, as well as potentially in the pathophysiology of disorders involving CNS myelination impairments, such as multiple sclerosis or schizophrenia (*Stedehouder and Kushner, 2017*; *Habermacher et al., 2019*). Therefore, it is important to determine the extent to which the morphological determinants of interneuron myelination are evolutionarily conserved. Here, we have performed axonal reconstructions and determined the topographical distribution of myelin internodes along human fast-spiking interneurons from acutely resected ex vivo neocortical tissue. We found that segmental neocortical interneuron myelination conforms to similar bivariate morphological thresholds as observed in mouse neocortex, on the basis of interbranch distance and axonal diameter, suggesting a species conservation of the biophysical constraints on the myelinating function of oligodendrocytes.

In conclusion, we demonstrate that the joint combination of interbranch distance and axonal shaft diameter accurately predicts the topography of neocortical interneuron myelination in vivo in both mouse and human.

## Materials and methods

### Mice

All experiments were approved by the Dutch Ethical Committee and in accordance with the Institutional Animal Care and Use Committee (IACUC) guidelines. The following mouse lines were obtained from Jackson Laboratory:

Pvalb$^{tm1(cre)Arbr}$/J mice (*Pvalb*::cre) (*Hippenmeyer et al., 2005*) www.jax.org/strain/008069
Sst$^{tm2.1(cre)Zjh}$/J (*Sst*::cre) (*Taniguchi et al., 2011*) (*Madisen et al., 2012*) www.jax.org/strain/013044
Gt(ROSA)26Sor$^{tm14(CAG-tdTomato)Hze}$/J (Ai14) www.jax.org/strain/013044 (*Madisen et al., 2012*)
Tsc1$^{tm1Djk}$/J (Tsc1$^{f/f}$) www.jax.org/strain/005680
C57BL/6J (WT) www.jax.org/strain/000664

Floxed *Ube3a* mice (*Ube3a$^{p+/mf}$*) were described previously (*Judson et al., 2016*).

All lines were backcrossed for more than 10 generations in C57BL/6J. Reporter lines were crossed to obtain heterozygous *Pvalb*::cre/heterozygous Ai14 (*Pvalb*::cre,Ai14) and heterozygous *Sst*:cre/heterozygous Ai14 (*Sst*::cre,Ai14). Mutant lines were crossed to obtain heterozygous *Pvalb*::cre/homozygous Tsc1$^{/f}$ (PV::TSC1). Heterozygous *Pvalb*::cre/heterozygous Ube3a$^{p+/mf}$ (PV::UBE3A) were obtained by crossing male homozygous *Pvalb*::cre with female Ube3a$^{p+/mf}$.

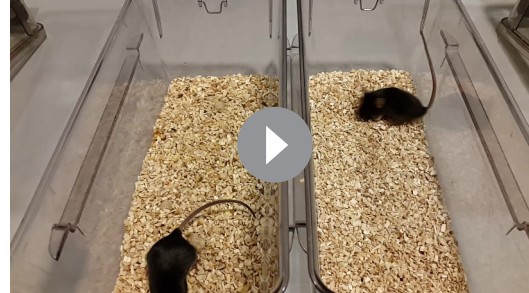

**Video 1.** Adult PV::TSC1 mice are ataxic. Left: PV::WT littermate. Right: PV::TSC1 mouse showing ataxia. https://elifesciences.org/articles/48615#video1

Heterozygous *Pvalb*::cre mice wildtype littermates for the mutant alleles from both breeding lines were used as controls (PV::WT). Both mutant lines were viable and healthy, although PV::TSC1 mice developed a severe ataxia during development (*Video 1*). We have occasionally observed spontaneous seizures during routine handling in PV::TSC1 but not PV::UBE3A lines. In addition, mutant lines were crossed to obtain heterozygous *Sst*::cre/homozygous *Tsc1^/f* (SOM::TSC1). Heterozygous *Sst*::cre mice wildtype littermates for the mutant alleles from this breeding lines were used as controls (SOM::WT). SOM::TSC1 mice did not display any behavioral abnormalities. No spontaneous seizures were observed during routine handling.

For all experiments, mice were used from 8 to 12 weeks of age. Mice were group-housed and maintained on a 12 hr light/dark cycle (lights on 07:00-19:00) with ad libitum access to food and water. All experiments were performed during the light phase of the cycle.

## Human brain tissue

Infiltrated peri-tumoral neocortical tissue was obtained from four patients undergoing tumor resection surgery at the Department of Neurosurgery (Erasmus University Medical Center, Rotterdam, The Netherlands). All procedures regarding human tissue were performed with the approval of the Medical Ethical Committee of the Erasmus University Medical Center. Written informed consent of each patient was provided in accordance with the Helsinki Declaration.

Patient #1 was an 84 year old male who presented with a glioblastoma in the right parieto-occipital lobe. He had no significant psychiatric or past medical history, and no history of epilepsy or seizures. Patient received no anti-epileptic or cytostatic medication.

Patient #2 was a 52 year old male who presented with metastases secondary to a melanoma in right temporal lobe. He had no significant psychiatric or other notable past medical history, and no history of epilepsy or seizures. Patient received no anti-epileptic or cytostatic medication.

Patient #3 was a 62 year old male who presented with a right temporal lobe glioblastoma. He had no history of seizures, or notable psychiatric or medical history. Patient received no anti-epileptic or cytostatic medication.

Patient #4 was a 80 year old male who presented with a glioblastoma in the right frontoparietal lobe. He had no significant psychiatric or other notable past medical history, no history of epilepsy or seizures, and received no anti-epileptic or cytostatic medication.

## Viral labeling and CNO injections

Adult mouse viral labelling was performed as reported before (*Stedehouder et al., 2017*), with minor adjustments. Specifically, adult PV::TSC1, PV::UBE3A, PV::WT, SOM::TSC1 and SOM::WT were used for axon diameter analysis, electrophysiology and single-cell reconstructions. Uncrossed heterozygous *Pvalb*::cre mice were used for PV-specific electron microscopy using APEX2.

The following viral vectors were used:

AAV9/CAG-Flex-eGFP (University of Pennsylvania Viral Vector Core).

AAV2/CAG-Flex-mGFP-APEX2 (a gift from ML Leyrer and DM Berson) (*Leyrer et al., 2016*).

Anesthesia was induced using 5% isoflurane ($O_2$ flow of 0.5 L/min), and subsequently maintained with 1–2% isoflurane during surgery. Body temperature was maintained at 37°C. Mice were placed into a custom-made stereotaxic frame using a mouth bar (Stoelting) for head fixation. Analgesia was provided systemically by subcutaneous Temgesic injection (buprenorphine 0.5 mg/kg) and locally by xylocaine spray (100 mg/mL, AstraZeneca) directly applied on the skull. To access the brain, a longitudinal scalp incision of ~1 cm length was made to reveal the skull, and a small craniotomy (<1 mm) was performed overlying the injection sites at the following coordinates (in mm): mPFC: +1.75 bregma, ±0.35 lateral, −1.9 dorsoventral (mm, from brain surface).

Mice used for electrophysiological recordings received 0.5 µL in a ¼ dilution in 0.1 M PB. Virus was aspirated in a borosilicate glass micropipette, which was slowly lowered to the target site. Virus injection was controlled by an automated syringe pump (infusion speed 0.1 µl/min). At the conclusion of the injection, the micropipette was maintained in place for 5 min and then slowly withdrawn. The surgical wound was closed with skin-glue (Derma+flex). Mice were left to recover for exactly 14 days to allow expression of the GFP protein. Importantly, mice were single-housed after surgery until used for electrophysiology or perfusion and immunofluorescence processing.

## Electrophysiology

### Mice

Anesthesia was induced using 5% isoflurane and mice were decapitated in ice-cold, NMDG-based cutting solution containing (in mM): 93 N-methyl-d-glucamine (NMDG), 93 HCl, 30 $NaHCO_3$, 25 D-glucose, 20 HEPES, 5 Na-ascorbate, two thiourea, 10 $MgCl_2$, 3 Na-pyruvate, 2.5 KCl, 1.25 $NaH_2PO_4$ and 0.5 $CaCl_2$ (300 mOsm, pH 7.4) oxygenated with 95% $O_2$/5% $CO_2$ before decapitation. After decapitation, the brain was quickly dissected. Coronal slices from the frontal cortex (300 μm) were cut with a vibrating slicer (Microm HM 650V, Thermo Scientific) and incubated in cutting solution at 37°C for 5 min., followed by oxygenated (95% $O_2$/5% $CO_2$) artificial cerebrospinal fluid (aCSF) at 37°C for 15 min. ACSF contained (in mM) 127 NaCl, 25 $NaHCO_3$, 25 D-glucose, 2.5 KCl, 1.25 $NaH_2PO_4$, 1.5 $MgSO_4$ and 1.6 $CaCl_2$. Slices were then allowed to recover at room temperature in the dark for at least 1 hr before recordings.

In *Pvalb*::cre;Ai14 and *Sst*::cre;Ai14 mice, PV+ and SOM+ interneurons respectively were visualized by native tdTomato fluorescence using an RFP filter (Semrock, Rochester, NY, USA). In PV:: TSC1, PV::WT, and PV::UBE3A mice, PV+ interneurons were visualized by expression of GFP using a GFP filter (Semrock, Rochester, NY, USA). Similarly, in SOM::TSC1 and SOM::WT mice, SOM+ interneurons were visualized by expression of eGFP using a GFP filter. Whole-cell recordings were made from layer III-V of the prelimbic area of the mPFC (between ~200 and 600 μm from midline; see *Figure 1—figure supplement 3*; bregma: +2.10 till +1.54 mm) at between ~20 μm and ~60 μm of the slice surface using borosilicate glass pipettes (3.5–5.5 MΩ resistance) with intracellular solution containing (in mM) 120 K-gluconate, 10 KCl, 10 HEPES, 10 K-phosphocreatine, 4 ATP-Mg, 0.4 GTP, and 5 mg/ml biocytin (pH was adjusted to 7.4 using KOH, and osmolarity measured 285–290 mOsm).

Recordings were performed in aCSF at near-physiological temperatures (32–33°C) using HEKA EPC10 quattro amplifiers and Patchmaster software (40 Hz sampling rate). Series resistance was typically <25 MΩ and fully compensated for bridge balance and capacitance; recordings in which the series resistance exceeded 25 MΩ were not included in the pooled averages. No correction was made for liquid junction potential. Data analysis was performed offline using Igor Pro v6 (Wavemetrics). Individual interneurons were recorded and filled for at least 20 min. In addition, cells were injected at least 10 times with large depolarizing currents in current clamp mode (500 pA, 5 Hz, 2 s) to facilitate biotin diffusion into fine axonal arbors.

Basic physiological characteristics were determined from voltage responses to square-wave current pulses of 500 ms duration, ranging from –100 pA to +400 pA, and delivered in 20 pA intervals. Input resistance was determined by the slope of the linear regression through the voltage-current curve. Sag was determined as the voltage difference between the lowest voltage response and the steady-state response at the last 50 ms to a square-wave current pulses of 500 ms duration at –100 pA. Single action potential (AP) characteristics were obtained from the first elicited action potential. AP threshold was defined as the inflection point at the foot of the regenerative upstroke. AP amplitude was defined as the voltage difference between the threshold and peak voltage. AP half-width was measured at half of the peak amplitude. AP rise time was quantified as duration from 10% to 90% of the peak amplitude. The fast after-hyperpolarizing potential (fAHP) amplitude was measured as the peak hyperpolarizing deflection from AP threshold following AP initiation. AP frequency was determined from the number of APs in response to a square-wave current pulse of 500 ms duration.

### Human

Ex vivo recordings of acutely resected frontal cortex were performed as previously described (*Stedehouder et al., 2017*). Non-eloquent overlaying tissue requiring surgical resection was utilized in order to access the location of a tumor. Immediately following resection, the tissue block was transferred to oxygenated (95% $O_2$/5% $CO_2$) ice-cold artificial cerebrospinal fluid (aCSF) containing (in mM) 127 NaCl, 25 $NaHCO_3$, 25 D-glucose, 2.5 KCl, 1.25 $NaH_2PO_4$, 1.5 $MgSO_4$ and 1.6 $CaCl_2$ during rapid transport to the laboratory. The time between surgical resection and tissue slicing was <10 min. Whole-cell recordings were performed similarly to mice as described above.

## Fluorescence immunohistochemistry

### Mouse

Deep anaesthesia was induced by intraperitoneal injection of pentobarbital, and mice were transcardially perfused with saline followed by 4% paraformaldehyde (PFA). Brains were dissected and post-fixed in 4% PFA for 2 hr at room temperature. Brains were transferred into 10% sucrose phosphate buffer (PB 0.1 M, pH 7.3) and stored overnight at 4°C. Embedding was performed in a 12% gelatin/10% sucrose block, with fixation in 10% paraformaldehyde/30% sucrose solution (PB 0.1 M) for 2 hr at room temperature and immersed in 30% sucrose (PB 0.1 M) at 4°C overnight. Forty micrometer coronal sections were collected serially (rostral to caudal) using a freezing microtome (Leica, Wetzlar, Germany; SM 2000R) and stored in 0.1 M PB. Sections were pre-incubated with a blocking PBS buffer containing 0.5% Triton X-100% and 10% normal horse serum (NHS; Invitrogen, Bleiswijk, The Netherlands) for 1 hr at room temperature. Sections were incubated in a mixture of primary antibodies in PBS buffer containing 0.4% Triton X-100% and 2% NHS for 72 hr at 4°C. The following primary antibodies were used:

> mouse anti-PV (1:1000, Swant, 235, lot #10-11(F)); RRID:AB_10000343
> rabbit anti-PV (1:1000, Swant PV25); RRID:AB_10000344
> goat anti-MBP (1:300, Santa Cruz, C-16, sc-13914, lot #F2416); RRID:AB_648798
> mouse anti-MBP (1:300, Santa Cruz, F-6, sc-271524); RRID:AB_10655672
> mouse anti-Ube3a (1:300, Sigma, 3E5, SAB1404508, lot #G4251-3E5); RRID:AB_10740376
> rabbit anti-pS6$^{S235/236}$ (1:300, Cell Signaling Technologies, 2211S, lot #23); RRID:AB_331679
> goat anti-SOM (1:300, Santa Cruz, D-20, sc-7819, lot #E1915); RRID:AB_2302603
> mouse anti-APC [CC1] (1:200, Abcam, ab16794); RRID:AB_443473

Sections were washed with PBS and incubated with corresponding Alexa-conjugated secondary antibodies (1:300, Invitrogen) and cyanine dyes (1:300, Sanbio, Uden, The Netherlands) in PBS buffer containing 0.4% Triton X-100, 2% NHS for 2–5 hr at room temperature. Sections were washed with PB 0.1 M and mounted on slides, cover slipped with Vectashield H1000 fluorescent mounting medium (Vector Labs, Peterborough, UK), sealed and imaged.

Recovery of biocytin-labelled cells following electrophysiological recordings was performed as reported before (*Stedehouder et al., 2017*), with minor alterations. Specifically, 300 µm slices were incubated overnight at 4°C in fresh 4% paraformaldehyde (PFA). Slices were extensively rinsed at room temperature in PBS and stained in PBS buffer containing 0.4% Triton X-100, 2% normal horse serum (NHS; Invitrogen, Bleiswijk, The Netherlands) and streptavidin-conjugated secondary antibody (1:300, Jackson; for PV::cre,Ai14-labelled and SOM::cre,Ai14-labelled cells) or streptavidin-Cy3 (1:300; Invitrogen; for GFP-labelled cells) overnight at 4°C. Slices were washed with PBS and PB 0.1 M and mounted on slides, cover slipped with 150 µl Mowiol (Sigma), sealed, and imaged for their axonal morphology (see Confocal imaging and analysis). To avoid excessive thinning or dehydration of 300 µm sections, cells were mounted, immediately imaged and returned to PB 0.1 M directly after imaging.

After full cell imaging, 300 µm slices were extensively washed in PB 0.1 M and incubated overnight at 4°C in 30% sucrose (0.1 M PB). Sections were then carefully recut at 40 µm using a freezing microtome (Leica, Wetzlar, Germany; SM 2000R) and stored serially in 0.1 M PB at 4°C. Serial 40 µm sections were extensively washed with PBS and pre-incubated with a blocking PBS buffer containing 0.5% Triton X-100% and 10% NHS for 1 hr at room temperature. Sections were incubated in PBS buffer containing 0.4% Triton X-100% and 2% NHS for 72 hr at 4°C and goat anti-MBP. Then, sections were washed with PBS, and incubated with corresponding Alexa-conjugated secondary antibodies (1:300, Invitrogen) and cyanine dyes (1:300, Sanbio, Uden, The Netherlands) in PBS buffer containing 0.4% Triton X-100, 2% NHS for 5 hr at room temperature. For biocytin, streptavidin-A488 (1:300, Jackson) and streptavidin-Cy3 (1:300, Invitrogen) were additionally used. Sections were washed with PB 0.1M and mounted on slides, cover slipped with Vectashield H1000 fluorescent mounting medium (Vector Labs, Peterborough, UK), sealed and imaged.

### Human tissue

Three-hundred µm electrophysiology slices with biocytin-labelled cells were incubated overnight at 4°C in 4% PFA. Slices were stained with secondary streptavidin-Cy3 (1:300, Sanbio, Uden, The Netherlands) in PBS buffer containing 0.5% Triton X-100% and 1% BSA for 5 hr at room temperature.

Next, three-hundred thick images were taken with confocal microscopy (see Confocal imaging and analysis). Slices were then rinsed at room temperature in 0.1 M PB, incubated for 16 hr at 4°C in 10% sucrose (0.1 M PB), and overnight at 4°C in 30% sucrose (0.1 M PB). Forty µm sections were collected serially using a freezing microtome (Leica, Wetzlar, Germany; SM 2000R) and stored in 0.1 M PB. Sections were extensively washed and pre-incubated with a blocking PBS buffer containing 0.5% Triton X-100% and 5% bovine serum albumin (BSA; Sigma-Aldrich, The Netherlands) for 1 hr at room temperature. Next, sections were incubated in a mixture of primary antibodies in PBS buffer containing 0.5% Triton X-100% and 1% BSA for 72 hr at 4°C. The following primary antibodies were used: mouse anti-MBP (1:300, Santa Cruz, F-6, sc-271524).

Sections were extensively washed with PBS (>2 hr), and incubated with corresponding Alexa-conjugated secondary antibodies (1:300, Invitrogen), and streptavidin-cyanine dyes (1:300, Sanbio, Uden, The Netherlands) in PBS buffer as previously described. Sections were washed with PB 0.1 M and mounted on slides, cover slipped with Vectashield H1000 fluorescent mounting medium (Vector Labs, Peterborough, UK) and sealed.

## Confocal imaging and analysis

Confocal imaging was performed using a Zeiss LSM 700 microscope (Carl Zeiss) equipped with Plan-Apochromat 10x/0.45 NA, 40x/1.3 NA (oil immersion) and 63x/1.4 NA (oil immersion) objectives. Alexa405/DAPI, Alexa488, Cy3/mCherry/tdTomato, and Alexa647 were imaged using excitation wavelengths of 405, 488, 555, and 639, respectively.

Quantification of interneuron-specific deletion of *Tsc1* or *Ube3a* as well as cell size analyses were performed in the prelimbic region in both hemispheres of the mPFC (bregma: +2.10 till +1.54 mm). We obtained tiled z-stack images (2048 × 2048 pixels) at 40x magnification with 1x digital zoom at a step size of 1 µm. Stacks were randomly sampled across layers II-V. For quantification of deletion efficiency, immunofluorescent somatic co-localization of PV or SOM and $pS6^{S235/236}$ or Ube3a was manually-counted using NIH ImageJ (version 1.41). At least three z-stacks were analyzed per mouse. For quantification of cell size, outlines of non-overlapping PV+ or SOM+ cell bodies were manually drawn and area and outline were calculated using Measure function of ImageJ. Cells were *post hoc* divided into pS6+ and pS6- or Ube3a+ and Ube3a- groups and compared.

Density of PV+, SOM+ and CC1+ cells were measured in the prelimbic area of the mPFC (bregma +2.20 till +1.70) in brain slices obtained from transcardially perfused animals. For PV+ cell counts, single plane 3 × 3 tile scan (1789 × 1789 µm) confocal images were obtained using the 10x objective with 1x digital zoom. All images captured both hemispheres. A 750 × 800 µm counting frame was established bilaterally from the midline. Within this counting frame, cells were manually counted utilizing the multi-point tool (Fiji image analysis software, version 2.0.0). Cell density was calculated in standardized $10^4$ µm$^2$ fields of view. SOM+ and CC1+ cell counts were performed using single plane 6 × 3 tile scan (1767 × 896 µm) confocal images capturing both hemispheres, with a 20x objective and 1x digital zoom.

Overall myelination in the prelimbic area of the mPFC (bregma +2.20 to +1.70) was measured in brain slices obtained from transcardially perfused animals. To capture both hemispheres, single plain, 6 × 3 tile scans (1767 × 896 µm) where made using a 20x objective with 1x digital zoom. MBP+ fluorescent area was quantified by employing the particle analysis tool in Fiji image analysis software (version 2.0.0).

Axonal examinations of virally-labelled cells were obtained by 63x magnification with 1x digital zoom and a step size of 0.5 µm. Cells were randomly sampled from layers II-V of the prelimbic area of the mPFC. Examination of occurrence of axonal myelination was performed offline using ImageJ. Axons were identified as the thinnest, smoothest, and most highly branched processes originating from either the soma or primary dendrite. In addition, axons seemed to branch at more obtuse ≥90° angles from one another, often turning back toward the soma, whereas dendrites branched at smaller angles (<90°), continuing their trajectory away from the soma. Additionally, SOM+ interneuron dendrites ubiquitously showed spines, whereas axons did not.

Axonal reconstructions of biocytin-filled cells were obtained at 63x magnification with 1x digital zoom and a step size of 0.5 µm. Images were transferred to Neurolucida 360 software (v2.8; MBF Bioscience) and reconstructed using interactive tracing with the Directional Kernels method. Reconstructed soma, axon and myelin segments were analyzed with Neurolucida Explorer (MBF Bioscience). All reconstructed PV+ cells had a classic basket cell morphology (*Jiang et al., 2015*), for

which none had a chandelier cell morphology. SOM+ cells predominantly exhibited a Martinotti morphology (*Jiang et al., 2015*) with an axonal arbor directed toward layer II-III. All SOM+ interneurons contained dendritic spines, whereas none of the PV+ interneurons did, including in any of the mutant lines.

Images for exact locations of all biocytin-filled cells (*Figure 1—figure supplement 3*) were obtained at 10x magnification with 1x digital zoom, and distance from the center of the soma till the midline was measured using ImageJ.

Axons were considered to be myelinated when they exhibited at least one MBP-positive myelinated internode. Axons were considered to be unmyelinated when we could not identify a single MBP-positive myelinated internode across the axon up to at least the 7th branch order in mice and 10th branch order in human cells. The distance to first myelin was defined as the distance along the axon from the soma, or in the case of dendrite-originating axons the distance from the originating dendrite, to the initial point of MBP immunofluorescence. Myelin segments that exited a slice were removed from subsequent analysis. Distance from internodes till consecutive branch points were quantified from the center of the branch point till the onset of MBP immunofluorescence. Internodes that were not followed by a branch point – followed by another internode or an axonal segment that exited the slice – were not taken along for analysis.

No spatial corrections were made for tissue shrinkage.

## Structured Illumination Microscopy (SIM) and Analysis

Imaging was performed using a Zeiss Elyra PS1 system. 3D-SIM data was acquired using a 63x/1.4 NA (oil immersion) objective. 488, 561 and 642 100 mW diode lasers were used to excite the fluorophores together with respectively a BP 495–575 + LP 750, BP 570–650 + LP 75 or LP 655 emission filter. For 3D-SIM imaging, a grating was present in the light path, modulated in 5 phases and five rotations, and multiple *z*-slices with an interval of 110 nm were recorded on an Andor iXon DU 885, $1002 \times 1004$ EMCCD camera. Raw images were reconstructed using Zen 2012 software (Zeiss), and analyzed with NIH ImageJ and Fiji image analysis software.

To avoid overexposure and induce background minimization, images were taken starting from the edge of the soma or dendrite following along the first axonal branch order onward. Each axonal segment was imaged individually from branch point till the next branch point, with the experimenter blinded to the myelination status of the segments. Axonal segments that exited the slice were removed from further analysis. Axonal segments that were predominantly oriented in the *z*-axis (maximum *z*-range: ~15–20 μm) were not taken along for further analysis.

For structured branch diameter analysis, images were loaded into Fiji and analyzed analogously to previously reported (*Chéreau et al., 2017*) using custom-written software. Briefly, an average-intensity projection was applied on individual axonal segments. A confocal whole-cell overview image and full reconstruction were used to track the centrifugal branch order of each traced segment. Segments were traced from the center of a branch point along the axon till the center of the next branch point using the Simple Neurite Tracer plugin for Fiji (*Longair et al., 2011*). Traces always followed the centrifugal direction away from the soma. Next, along the trace, perpendicular lines of 50 pixels (equals 2 μm) were placed on every pixel (~40 nm) along the trace. On these perpendicular lines, biotin fluorescence intensity values were determined and a Gaussian curve was fitted on the intensity profile. Only fits with $r^2 > 0.9$ were included in further analysis, resulting in a loss of approximately ~10% of axonal pixel measurements. Subthreshold fits ($r^2 < 0.9$) occurred due to occasional high background fluorescence, other processes in close proximity (axons/dendrites), extensive axon curvature, or rare axonal filopodia. Next, from the Gaussian fit the full width at half maximum (FWHM) was calculated. Consecutive FHWM values were not averaged, and individual pixel FWHM values are provided. Diameter profiles of full axon segments (excluding branch points and *en passant* boutons) were generated and analyzed further. Axonal segments that exited the slice before reaching the next branch point would be excluded from further analysis. Where applicable, after complete analysis segments were divided into unmyelinated and myelinated segments.

For comparative axon diameter analysis of PV+ and SOM+ interneurons in PV::TSC1, PV::UBE3A, PV::WT, SOM::TSC1 and SOM::WT, GFP-transduced cells from layers II till V were imaged, and the axon diameter of individual cells was determined at the start of the AIS at ~3–5 μm from the soma or from the originating dendrite using FWHM measurements in Fiji. Similar to the detailed axon segment reconstructions, only measurements with fits > 0.9 were included in the analysis.

## mGFP-APEX2 labeling, Electron Microscopy and image analysis

Adult heterozygous *Pvalb*::cre mice were unilaterally injected (see **Viral Labeling**) with AAV2/CAG-Flex-mGFP-APEX2 in a 0.5 µl bolus undiluted with a titre of $2.93 \times 10^{12}$ in the adult mPFC using the following coordinates: mPFC: +1.75 from bregma,±0.35 lateral, −1.9 dorsoventral (mm, from brain surface). After 7–14 days, mice were anesthetized by intraperitoneal injection of pentobarbital and transcardially perfused with saline followed by ice-cold 4% paraformaldehyde (PFA)/1% glutaraldehyde in 0.1M PB. Brains were carefully dissected and post-fixed in the same solution overnight at 4° C. The brains were washed extensively in cold 0.1 M PB, and 100 µm coronal slices from the frontal cortex (bregma: +2.10 till +1.54 mm) were cut on a vibrating slicer (Microm HM 650V, Thermo Scientific). Sections were serially stored in cold 0.1M PB and processed for 3–3'-diaminobenzidine (DAB) staining.

Sections were incubated in full concentration DAB (0.1 M PB, 0.66% DAB, 0.033% $H_20_2$) for 6 hr at room temperature in the dark. Sections were washed in cold PB and post-fixed in cold 2% glutaraldehyde in 0.1 M PB for 2 hr at 4°C. Regions of interest were manually cut out and processed for electron microscopy. Samples were post-fixed in 1% osmium tetraoxide, dehydrated, and embedded in epoxy resin. Ultrathin sections (40–60 nm) were cut on a Leica Ultramicrotome Supercut UCT, contrasted with uranyl acetate and lead citrate and analyzed in a Phillips CM100 electron microscopy (Aachen, Germany) at 80 kV. Multiple non-overlapping regions were imaged at 14kx and analyzed off-line using Fiji image software.

## Statistical analysis

Statistical analysis was performed using IBM SPSS (version 23). Data sets were analyzed using Shapiro-Wilk test for normality. No outlier data were identified or removed. Experiments were designed using sample sizes comparable to previously published studies (*Stedehouder et al., 2017*; *Stedehouder et al., 2018*). Masking was used for group allocation, data collection, and data analysis whenever possible. Data sets with normal distributions were analyzed for significance using unpaired Student's two-tailed *t*-test or analysis of variance (ANOVA) measures followed by Tukey's *post hoc* test. Data are expressed as mean ± standard error. Data sets with non-normal distributions were analyzed using Mann-Whitney *U* test or Kruskal-Wallis test with Dunn's adjustment for multiple comparisons.

Receiver operating characteristic (ROC) curves were generated using a custom-written algorithm to implement univariate (*Hanley and McNeil, 1982*) and bivariate (*Wang and Li, 2015*) methods. Univariate and bivariate thresholds were determined at the corresponding points of maximization of the Youden's *J* statistic (*Jin and Lu, 2009*), represented as the sum of the sensitivity and specificity. Area under the curve (AUC) values were computed as the integral of the univariate or bivariate ROC curves.

Exact *P*-values values are provided in the text, except when p<0.001. Significance threshold was set at p<0.05.

## Acknowledgements

We thank the patients for their participation in this study, and the Erasmus MC neurosurgical team for facilitating the contribution of the tissue samples. Funding for this project was provided by the European Commission [NEURON-JTC2018-024, ERA-PerMed2018-127, H2020-FETPROACT-824070], Netherlands Organisation for Health Research and Development (ZonMW) [013.18.002, 40-00812-98-15030, 456.008.003], Netherlands Organization for Scientific Research (NWO) [NOCI Zwaartekracht 024.003.001], and Erasmus MC Desiderius Award to SAK, Erasmus MC Fellowship to ZG.

## Additional information

### Funding

| Funder | Grant reference number | Author |
| --- | --- | --- |
| Horizon 2020 Framework Programme | NEURON-JTC2018-024 | Steven A Kushner |
| ZonMw | 40-00812-98-15030 | Steven A Kushner |
| Nederlandse Organisatie voor Wetenschappelijk Onderzoek | 834.12.002 | Steven A Kushner |
| European Commission | ERA-PerMed2018-127 | Steven A Kushner |
| European Commission | H2020-FETPROACT-824070 | Steven A Kushner |
| ZonMw | 013.18.002 | Steven A Kushner |
| ZonMw | 456.008.003 | Steven A Kushner |
| Erasmus Medisch Centrum | Desiderius Award | Steven A Kushner |
| Erasmus Medisch Centrum | Fellowship | Zhenyu Gao |

The funders had no role in study design, data collection and interpretation, or the decision to submit the work for publication.

### Author contributions

Jeffrey Stedehouder, Conceptualization, Formal analysis, Investigation, Methodology, Writing—original draft, Writing—review and editing; Demi Brizee, Formal analysis, Validation, Investigation, Methodology, Writing—review and editing; Johan A Slotman, Software, Formal analysis, Methodology, Writing—review and editing; Maria Pascual-Garcia, Validation, Investigation, Writing—review and editing; Megan L Leyrer, Bibi LJ Bouwen, David M Berson, Resources, Methodology, Writing—review and editing; Clemens MF Dirven, Zhenyu Gao, Resources, Supervision, Writing—review and editing; Adriaan B Houtsmuller, Methodology, Writing—review and editing; Steven A Kushner, Conceptualization, Formal analysis, Supervision, Funding acquisition, Writing—original draft, Writing—review and editing

### Author ORCIDs

Jeffrey Stedehouder https://orcid.org/0000-0001-6017-7279
Johan A Slotman http://orcid.org/0000-0001-9705-9620
Megan L Leyrer http://orcid.org/0000-0001-5670-4005
Zhenyu Gao http://orcid.org/0000-0002-4979-2366
Steven A Kushner https://orcid.org/0000-0002-9777-3338

### Ethics

Human subjects: All procedures regarding human tissue were performed with the approval of the Medical Ethical Committee of the Erasmus University Medical Center. Written informed consent of each patient was provided in accordance with the Helsinki Declaration.
Animal experimentation: This study was performed in strict accordance with the recommendations in the Guide for the Care and Use of Laboratory Animals of the National Institutes of Health. All of the animals were handled according to approved institutional animal care and use committee (IACUC) protocols (IG 15-064) of the Dutch Ethical Committee (DEC). The protocol was approved by the Netherlands Centrale Commissie Dierproeven (Permit Number: AVD1010020173544). All surgery was performed under isoflurane anesthesia, and every effort was made to minimize suffering.

### Decision letter and Author response

Decision letter https://doi.org/10.7554/eLife.48615.sa1
Author response https://doi.org/10.7554/eLife.48615.sa2

## Additional files

### Supplementary files

• Source code 1. Fiji source code for automated quantification of axonal diameter within user-defined segments based on the Gaussian full-width at half-maximum of the orthogonal cross-section of fluorescence intensity.

• Supplementary file 1. Electrophysiological properties of *Pvalb*::cre,Ai14 PV+ cells.

• Supplementary file 2. Electrophysiological properties of *Sst*::cre,Ai14 SOM+ cells.

• Supplementary file 3. Electrophysiological properties of human fast-spiking interneurons.

• Transparent reporting form

### Data availability

All data generated or analysed during this study are included in the manuscript and supporting files. Source data files have been provided for Figures 1–9. Fiji ImageJ code for semi-automated reconstruction of axon diameter has been provided as an additional file. Full human FS and mouse PV cell reconstructions are available at NeuroMorpho (http://doi.org/10.13021/EXNK-G157) [Cell IDs: Human: 181002.01, 161012.08; Mouse: 170113.01, 170316.02, 170426.02].

The following dataset was generated:

| Author(s) | Year | Dataset title | Dataset URL | Database and Identifier |
|---|---|---|---|---|
| Jeffrey Stedehouder, Demi Brizee, Steven A Kushner | 2019 | Full dendritic and axonal reconstructions of human and mouse neocortical layer 3 fast-spiking interneurons (Human: right occipital cortex [161012.08], right temporal cortex [181002.01]; Mouse: medial prefrontal cortex [170113.01, 170316.02, 170426.02]) | http://doi.org/10.13021/EXNK-G157 | NeuroMorpho, 10.13021/EXNK-G157 |

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
