## [Decision Letter]

Thank you for submitting your article "Local axonal morphology guides the topography of interneuron myelination in mouse and human neocortex" for consideration by *eLife*. Your article has been reviewed by three peer reviewers, including Sacha B Nelson as the Reviewing Editor and Reviewer #1, and the evaluation has been overseen by a Reviewing Editor and Eve Marder as the Senior Editor. The following individuals involved in review of your submission have agreed to reveal their identity: Maria Cecilia Angulo (Reviewer #2).

The reviewers have discussed the reviews with one another and the Reviewing Editor has drafted this decision to help you prepare a revised submission.

Summary:

The authors use structured illumination microscopy and EM to visualize myelinated axons of PV+ GABAergic interneurons in prefrontal cortex of the mouse and in ex vivo tissue samples from human patients. Following up on their prior work showing these neurons are frequently myelinated, the authors develop a bivariate model that accurately predicts the dependence of myelination on interbranch distance and axonal diameter. They test the model by genetically altering axon caliber using mutations of *Tsc1* and *Ube3a* both in the PV+ neurons and in a second population of interneurons (SST+) which normally have reduced myelination.

This study is well-performed, very interesting and timely. It tackles an important question in an emerging topic in the field of oligodendrocytes which concerns the mechanisms governing the myelination of local GABAergic interneurons. The measurements appear to have been carefully made and are well illustrated. The bivariate model is convincingly improved relative to univariate models of branch distance or diameter alone. Nevertheless, there are some points that need to be addressed to support the conclusions.

Essential revisions:

1) The modifications in the myelination of PV and SST interneurons should be specific for each cell type in the different knockout mice since these cells were targeted using PV-Cre and SST-Cre. It would be important to verify two points: (a) is there an impact on the global pattern of myelination in the PFC in the different knockout mice? (b) Are these differences attributable specifically to myelination changes in PV or SST cells (according to the knockout mouse). Figure 8D shows a strong increase in the general myelination of SOM::TSC1 mice compared to controls. This increase seems high compared to the proportions of both SST interneurons and myelinated GABAergic interneurons in the PFC (Stedehouder et al., 2017). Authors did not show similar images for PV::TSC1 and PV::UBE3A. In the same line, is there any modification in the cell density of PV and SST interneurons in the different knockout mice that could explain changes in the general myelination? The authors already have the immunostainings and they should be able to answer these points with their already processed material.

2) A related question is: Does cell-specific deletion of *Tsc1* or *Ube3a* have direct or indirect effects on myelination independent of changes in axon caliber and cell soma size?a) Changes in the pattern and extent of myelination of GABAergic axons require changes to oligodendrocytes. Do these cell-specific deletions effect numbers of oligodendrocyte precursors and mature oligodendrocytes? Figure 8D suggests the answer maybe yes. The current model the authors propose does not include these parameters, which are essential to determining whether changes in axonal morphology that are driving these changes in myelination pattern or other effects that are contributing to this phenotype. Thus, the claim that axonal morphology is necessary and sufficient to govern GABAergic myelination is too strong.b) Is there a cell-autonomous effect of gene deletions on axonal myelination? Non-recombined cells may provide insights into this question.

3) Since the axon morphology and size of the cells are modified in knockout mice, we would expect drastic changes in the intrinsic electrophysiological properties of the target cells. This is not the case, in particular for PV neurons. Since no changes were observed for the different studied parameters, the authors could compare the distribution of their data rather than the means (for instance, Figure 4—figure supplement 4F seems to display different distributions). It would be also important to analyze other parameters important for PV cells such as the amplitude of the second spike, the duration increase and amplitude reduction between the first and second APs, the early and late accommodation of AP discharges. Did the capacitance of PV and SST cells changed in correlation with the morphological modifications of the soma and axons? The authors already performed the recordings and should be able to do this analysis.

4) Does the diameter/morphology of GABAergic axons increase with maturity and subsequently affect the relationship with myelination?

5) It is nice the authors managed to work in human tissue. However, it seems from their data of Figure 9J and K that the effect of axonal diameter and interbranch distances is less strong in human than in mouse PV interneurons. The authors might amend their conclusions on the similarities of both species throughout the manuscript, being more cautious.

---

## [Author Response]

Essential revisions:1) The modifications in the myelination of PV and SST interneurons should be specific for each cell type in the different knockout mice since these cells were targeted using PV-Cre and SST-Cre. It would be important to verify two points: (a) is there an impact on the global pattern of myelination in the PFC in the different knockout mice? (b) Are these differences attributable specifically to myelination changes in PV or SST cells (according to the knockout mouse). Figure 8D shows a strong increase in the general myelination of SOM::TSC1 mice compared to controls. This increase seems high compared to the proportions of both SST interneurons and myelinated GABAergic interneurons in the PFC (Stedehouder et al., 2017). Authors did not show similar images for PV::TSC1 and PV::UBE3A. In the same line, is there any modification in the cell density of PV and SST interneurons in the different knockout mice that could explain changes in the general myelination? The authors already have the immunostainings and they should be able to answer these points with their already processed material.

We agree that these are important questions to be addressed. Therefore, we have now quantified total myelin content across each of the mouse lines examined, which has been added as a new Figure 5—figure supplement 1. Across the different PV-Cre knockout lines, we found no discernible changes in MBP labeling. In contrast, SST::TSC1 mice exhibit a robust increase in MBP labeling compared to littermate controls (updated Figure 8D and E). We have now modified the Results section to reflect the quantification of global MBP labeling across the different knockout lines:

Subsection “Bi-directional manipulation of PV+ axon morphology alters myelination”: “Despite these PV cell type-specific alterations in myelination, no robust changes were observed in global myelination (Figure—figure supplement 1A and B) or CC1+ mature oligodendrocyte density (Figure 5—figure supplement 1C and D).”

Subsection “Genetic manipulation of SOM+ axon morphology induces de novo myelination”: “Consistent with the high proportion of SOM+ cell myelination, the mPFC of SOM::TSC1 exhibited a marked increase of global myelination (Figure 8D and E) and a higher density of CC1+ mature oligodendrocytes (Figure 8F and G).”

In addition, we have also performed the requested cell counts across the different knockout lines in the PFC. No significant changes of PV (Figure 4—figure supplement 3) or SST (Figure 7—figure supplement 2) interneuron density were observed. These findings have been added to the revised manuscript in the Results section:

Subsection “Bi-directional manipulation of PV+ axon morphology alters myelination”: “Notably, mPFC PV cell density was similar across PV::UBE3A, PV::TSC1, and PV::WT mice (Figure 4—figure supplement 3A and B).”

Subsection “Genetic manipulation of SOM+ axon morphology induces de novo myelination”: “SOM^+^ cell density was unchanged in the mPFC of SOM::TSC1 mice (Figure 7—figure supplement 2A and B).”

We acknowledge that this does still leave open the question of whether the increase of myelination observed in the SST::TSC1 mice is specifically restricted to myelination along axons of SST+ cells, or whether an increase in myelination might also occur on other cell types. However, we hope the reviewers agree that definitively addressing this issue would require a considerable amount of additional work to answer a question that is beyond the central focus of this manuscript. Moreover, we have been very careful not to make any claims in the manuscript about potential non-cell autonomous effects of deletion of *Tsc1* in SST+ cells and explicitly acknowledge this possibility in the Discussion section:

“However, compared to the PV::TSC1 cell manipulation, SOM::TSC1 mice exhibited a more robust increase of global myelination in the mPFC, which was accompanied by an increased number of CC1+ mature oligodendrocytes. […] A second non-mutually exclusive possibility is that SOM::TSC1 deletion results in additional non-cell autonomous effects that enhance global myelination and recruitment of mature oligodendrocytes.”.

2) A related question is: Does cell-specific deletion of Tsc1 or Ube3a have direct or indirect effects on myelination independent of changes in axon caliber and cell soma size?a) Changes in the pattern and extent of myelination of GABAergic axons require changes to oligodendrocytes. Do these cell-specific deletions effect numbers of oligodendrocyte precursors and mature oligodendrocytes? Figure 8D suggests the answer maybe yes. The current model the authors propose does not include these parameters, which are essential to determining whether changes in axonal morphology that are driving these changes in myelination pattern or other effects that are contributing to this phenotype. Thus, the claim that axonal morphology is necessary and sufficient to govern GABAergic myelination is too strong.

We agree that an examination of the impact of the cell-type specific deletions on oligodendrocyte lineage cells would provide a more comprehensive picture. Accordingly, we have now performed quantification of mature oligodendrocytes across each of the different knockout lines. Consistent with the results of global MBP labeling, we found no differences across the PV mutant lines (Figure 5—figure supplement 1), but a significant increase in oligodendrocytes in the PFC of the SOM::TSC1 mice (Updated Figure 8F and G). These findings have been added to the revised manuscript in the Results section:

Subsection “Bi-directional manipulation of PV+ axon morphology alters myelination”: “Despite these PV cell type-specific alterations in myelination, no robust changes were observed in global myelination (Figure 5—figure supplement 1A and B) or CC1+ mature oligodendrocyte density (Figure 5—figure supplement 1C and D).”

Subsection “Genetic manipulation of SOM+ axon morphology induces de novo myelination”: “Consistent with the high proportion of SOM+ cell myelination, the mPFC of SOM::TSC1 exhibited a marked increase of global myelination (Figure 8D and E) and a higher density of CC1+ mature oligodendrocytes (Figure 8F and G).”

We have also moderated the “necessary and sufficient” claim in the revised text.

b) Is there a cell-autonomous effect of gene deletions on axonal myelination? Non-recombined cells may provide insights into this question.

We have attempted to answer this question with two independent experiments:

i) Cre-dependent eGFP virus injections into GAD2-cre x *Tsc1^f/f^* and WT littermate mice, followed by somatostatin, pS6^235/236^, and MBP immunofluorescence to examine axonal myelination of non-recombined PV+ and SOM+ cells.

ii) We have performed triple immunofluorescent staining for somatostatin, NF200 (axonal marker) and MBP in PV-cre x *Tsc1^f/f^* and WT mice.

Unfortunately, neither experiment allowed for sufficiently high-resolution quantification of myelination of non-recombination cells. Due to the relatively small proportion of nonrecombined cells, addressing this question would require breeding triple transgenic mice (e.g., SOM-cre x Cre-dependent fluorescent reporter x *Tsc1^f/f^*), followed by two independent groups of biocytin labeling/reconstruction for recombined and non-recombined cells, for which the latter group would be very low yield since the rate of non-recombination is small and only definitively visualizable by post-hoc immunohistochemistry, thereby precluding targeted biocytin filling of the non-recombined population.

3) Since the axon morphology and size of the cells are modified in knockout mice, we would expect drastic changes in the intrinsic electrophysiological properties of the target cells. This is not the case, in particular for PV neurons. Since no changes were observed for the different studied parameters, the authors could compare the distribution of their data rather than the means (for instance, Figure 4—figure supplement 4F seems to display different distributions). It would be also important to analyze other parameters important for PV cells such as the amplitude of the second spike, the duration increase and amplitude reduction between the first and second APs, the early and late accommodation of AP discharges. Did the capacitance of PV and SST cells changed in correlation with the morphological modifications of the soma and axons? The authors already performed the recordings and should be able to do this analysis.

We would also have expected to find changes in electrophysiological properties given the morphological alterations. The reason for the overall limited effect is unknown, however likely to be due to homeostatic compensation due to the early neurodevelopmental deletion. With regard to the additional requested analyses, regrettably our recordings neglected to allow for measurement of cell capacitance. However, as suggested by the Reviewers, we have additionally analyzed the distributions for all parameters, but have not uncovered any significant differences. In addition, we have analyzed the second AP for all datasets, which also did not reveal differences between groups.

4) Does the diameter/morphology of GABAergic axons increase with maturity and subsequently affect the relationship with myelination?

This is a fascinating, important and largely unexplored question. However, we hope that the reviewers and editors would agree that this question falls outside the scope of the current manuscript.

5) It is nice the authors managed to work in human tissue. However, it seems from their data of Figure 9J and K that the effect of axonal diameter and interbranch distances is less strong in human than in mouse PV interneurons. The authors might amend their conclusions on the similarities of both species throughout the manuscript, being more cautious.

We agree that the optimal ROC model fit, while still quite robust, appears less strong in human compared to mouse. In an effort to improve the statistical power of the model fit, we now succeeded to reconstruct another human fast-spiking interneuron from an additional patient, which has further strengthened our results. Nevertheless, we have moderated the strength of our conclusions regarding the robustness of the model fit for the human cells, and the correspondence between the mouse and human data, in the Introduction, Results section and in the Discussion section.